# An Adaptive Dual-Population Collaborative Chicken Swarm Optimization Algorithm for High-Dimensional Optimization

**DOI:** 10.3390/biomimetics8020210

**Published:** 2023-05-19

**Authors:** Jianhui Liang, Lifang Wang, Miao Ma

**Affiliations:** 1School of Computer Science, Northwestern Polytechnical University, Xi’an 710072, China; 993952@hainanu.edu.cn; 2School of Applied Science and Technology, Hainan University, Danzhou 571737, China; 3School of Computer Science, Shaanxi Normal University, Xi’an 710062, China

**Keywords:** meta-heuristic optimization, chicken swarm optimization, high-dimensional optimization

## Abstract

With the development of science and technology, many optimization problems in real life have developed into high-dimensional optimization problems. The meta-heuristic optimization algorithm is regarded as an effective method to solve high-dimensional optimization problems. However, considering that traditional meta-heuristic optimization algorithms generally have problems such as low solution accuracy and slow convergence speed when solving high-dimensional optimization problems, an adaptive dual-population collaborative chicken swarm optimization (ADPCCSO) algorithm is proposed in this paper, which provides a new idea for solving high-dimensional optimization problems. First, in order to balance the algorithm’s search abilities in terms of breadth and depth, the value of parameter *G* is given by an adaptive dynamic adjustment method. Second, in this paper, a foraging-behavior-improvement strategy is utilized to improve the algorithm’s solution accuracy and depth-optimization ability. Third, the artificial fish swarm algorithm (AFSA) is introduced to construct a dual-population collaborative optimization strategy based on chicken swarms and artificial fish swarms, so as to improve the algorithm’s ability to jump out of local extrema. The simulation experiments on the 17 benchmark functions preliminarily show that the ADPCCSO algorithm is superior to some swarm-intelligence algorithms such as the artificial fish swarm algorithm (AFSA), the artificial bee colony (ABC) algorithm, and the particle swarm optimization (PSO) algorithm in terms of solution accuracy and convergence performance. In addition, the APDCCSO algorithm is also utilized in the parameter estimation problem of the Richards model to further verify its performance.

## 1. Introduction

High-dimensional optimization problems generally refer to ones with high complexity and dimensions (exceeding 100). It often has the characteristics of non-linearity and high complexity. In real life, many problems can be expressed as high-dimensional optimization problems, such as large-scale job-shop-scheduling problems [1], vehicle-routing problems [2], feature selection [3], satellite autonomous observation mission planning [4], economic environmental dispatch [5], and parameter estimation. These kinds of optimization problems often greatly degrade the performance of the optimization algorithm as the dimension of the optimization problem increases, so it is extremely difficult to obtain the global optimal solution, which poses a technical challenge to solving many practical problems. Therefore, the study of high-dimensional optimization problems has important theoretical and practical significance [6,7].

The meta-heuristic optimization algorithm is a class of random search algorithms proposed by simulating biological intelligence in nature [8], and has been successfully applied in various fields, such as the Internet of Things [9], network information systems [10,11], multi-robot space exploration [12], and so on. At present, hundreds of algorithms have emerged, such as the particle swarm optimization (PSO) algorithm, the artificial bee colony (ABC) algorithm, the artificial fish swarm algorithm (AFSA), the bacterial foraging algorithm (BFA), the grey wolf optimizer (GWO) algorithm, and the sine cosine algorithm (SCA) [13]. These algorithms have become effective methods for solving high-dimensional optimization problems because of their simple structure and strong exploration and exploitation abilities. For example, Huang et al. proposed a hybrid optimization algorithm by combining the frog’s leaping optimization algorithm with the GWO algorithm and verified the performance of the algorithm on 10 high-dimensional complex functions [14]. Gu et al. proposed a hybrid genetic grey wolf algorithm for solving high-dimensional complex functions by combining the genetic algorithm and GWO and verified the performance of the algorithm on 10 high-dimensional complex test functions and 13 standard test functions [15]. Wang et al. improved the grasshopper optimization algorithm by introducing nonlinear inertia weight and used it to solve the optimization problem of high-dimensional complex functions. Experiments on nine benchmark test functions show that the algorithm has significantly improved convergence speed and convergence accuracy [16].

The chicken swarm optimization (CSO) algorithm is a meta-heuristic optimization algorithm proposed by Meng et al. in 2014, which simulates the foraging behavior of chickens in nature [17]. The algorithm realizes rapid optimization through information interaction and collaborative sharing among roosters, hens, and chicks. Because of its good solution accuracy and robustness, it has been widely used in network engineering [18,19], image processing [20,21,22], power systems [23,24], parameter estimation [25,26], and other fields. For example, Kumar et al. utilized the CSO algorithm to select the best peer in the P2P network and proposed an optimal load-balancing strategy. The experimental results show that it has better load balancing than other methods [18]. Cristin et al. applied the CSO algorithm to classify brain tumor severity in magnetic resonance imaging (MRI) images and proposed a brain-tumor image-classification method based on the fractional CSO algorithm. Experimental results show that this method has good performance in accuracy, sensitivity, and so on [20]. Liu et al. developed an improved CSO–extreme-learning machine model by improving the CSO algorithm and applied it to predict the photovoltaic power of a power system and obtained satisfactory results [23]. Alisan applied the CSO algorithm for the parameter estimation of the proton exchange membrane fuel cell model, and it exhibit particularly good performance [25].

Although the CSO algorithm has been successfully applied to various fields and solved many practical problems, the above application examples are all aimed at low-dimensional optimization problems. With the increase in the dimensions of the optimization problems, the CSO algorithm is prone to premature convergence. Therefore, for the optimization problem of high-dimensional complex functions, Yang et al. constructed a genetic CSO algorithm by introducing the idea of a genetic algorithm into the CSO algorithm and verified the performance of the proposed algorithm on 10 benchmark functions [27]. Although the convergence speed and stability were improved, the solution accuracy is still unsatisfactory. Gu et al. realized the solution to high-dimensional complex function optimization problems by removing the chicks in the chicken swarm and introducing an inverted S-shaped inertial weight to construct an adaptive simplified CSO algorithm [28]. Although the proposed algorithm is significantly better than some other algorithms in solution accuracy, there is still room for improvement in convergence speed. By introducing the dissipative structure and differential mutation operation into the basic CSO algorithm, Han constructed a hybrid CSO algorithm to avoid premature convergence in solving high-dimensional complex problems, and verified the performance of the proposed algorithm on 18 standard functions [29]. Although its convergence performance was improved, the solution accuracy should be further enhanced.

To address the aforementioned issues, we propose an adaptive dual-population collaborative CSO (ADPCCSO) algorithm in this paper. The algorithm solves high-dimensional complex problems by using an adaptive adjustment strategy for parameter *G*, an improvement strategy for foraging behaviors, and a dual-population collaborative optimization strategy. Specifically, the main technical features and originality of this paper are given below.

(1) The value of parameter *G* is given using an adaptive dynamic adjustment method, so as to balance the breadth and depth of the search abilities of the algorithm.

(2) To improve the solution accuracy and depth optimization ability of the CSO algorithm, an improvement strategy for foraging behaviors is proposed by introducing an improvement factor and adding a kind of chick’s foraging behavior near the optimal value.

(3) A dual-population collaborative optimization strategy based on the chicken swarm and artificial fish swarm is constructed to enhance the global search ability of the whole algorithm.

The simulation experiments on the selected standard test functions and the parameter estimation problem of the Richards model show that the ADPCCSO algorithm is better than some other meta-heuristic optimization algorithms in terms of solution accuracy, convergence performance, etc.

The rest of this paper is arranged as follows. In Section 2, the principle and characteristics of the standard CSO algorithm are briefly introduced. Section 3 describes the ADPCCSO algorithm proposed in this paper in detail, the improvement strategies of the algorithm, and the main implementation steps are presented in this section. Simulation experiments and analysis are presented in Section 4 to verify the performance of the proposed ADPCCSO algorithm. Finally, we conclude the paper in Section 5.

## 2. The Basic CSO Algorithm

CSO algorithm is a class of random search algorithm based on the collective intelligent behavior of chicken swarms in the process of foraging. In this algorithm, several randomly generated positions in the search range are regarded as several chickens, and the fitness function values of chickens are regarded as food sources. In light of the fitness function values, the whole chicken swarm is divided into the roosters, hens, and chicks, where roosters have the best fitness values, hens take second place, and chicks have the worst fitness values. The algorithm relies on the roosters, hens, and chicks to constantly conduct information interaction and cooperation sharing and finally finds the best food source [30,31]. The characteristics are as follows:

(1) The whole chicken swarm is divided into several subgroups, and each subgroup is composed of a rooster, at least one hen and several chicks. The hens and chicks look for food under the leadership of the roosters in their subgroups, and they will also obtain food from other subgroups.

(2) In the basic CSO algorithm, once the hierarchical relationship and dominance relationship between roosters, hens, and chicks are determined, they will remain unchanged for a certain period until the role update condition is met. In this way, they achieve information interaction and find the best food source.

(3) The whole algorithm realizes parallel optimization through the cooperation between roosters, hens, and chicks. The formulas corresponding to their foraging behaviors are as follows:

The roosters’ foraging behavior:(1)$$X_{i,j}\left(t+1\right)=X_{i,j}\left(t\right)\times \left(1+Randn\left(0,\sigma^{2}\right)\right)\;j\in (1,2,\;\ldots \;Dim)$$
(2)$$\sigma^{2}=\left\{\begin{array}{l} 1,\;f_{i}\leq f_{k,}\\ \exp\left(\frac{\left(f_{k}\;-\;f_{i}\right)}{\left|f_{i}\right|\;+\;\varepsilon }\right),f_{i}>f_{k} \end{array}\right.\;k\neq i$$
where $X_{i,j}\left(t\right)$ stands for the position of the *i*th rooster at iteration *t*. *Dim* is the dimension of the problem to be solved. $Randn\left(0,\sigma^{2}\right)$ is a random number matrix with a mean value of 0 and a variance of $\sigma^{2}$. ε is a smallest positive normalized floating-point number in IEEE double precision. $f_{k}$ is the fitness function value of any rooster, and k≠i.

The hens’ foraging behavior is described by
(3)$$X_{i,j}\left(t+1\right)=X_{i,j}\left(t\right)+c_{1}\times rand()\times \left(X_{r_{1},j}\left(t\right)-X_{i,j}\left(t\right)\right)+c_{2}\times rand()\times \left(X_{r_{2},j}\left(t\right)-X_{i,j}\left(t\right)\right)$$
(4)$$c_{1}=\exp\left(\left(f_{i}-f_{r_{1}}\right)/\left(abs\left(f_{i}\right)+\varepsilon \right)\right)$$
(5)$$c_{2}=\exp\left(f_{r_{2}}-f_{i}\right)$$
where $X_{i,j}\left(t\right)$ is the individual position of the *i*th hen, $X_{r1,j}\left(t\right)$ is the position of the group-mate rooster of the *i*th hen, $X_{r2,j}\left(t\right)$ is a randomly selected chicken, and $r_{2}\neq r_{1}$.

The chicks’ foraging behavior is described by
(6)$$X_{i,j}\left(t+1\right)=X_{i,j}\left(t\right)+FL\times \left(X_{m,j}\left(t\right)-X_{i,j}\left(t\right)\right)$$
where *i* is an index of the chick, and *m* is an index of the *i*th chick’s mother. $FL\in \left(0,2\right)$ is a follow coefficient.

## 3. ADPCCSO Algorithm

To address the issue of precocious convergence of the basic CSO algorithm in solving high-dimensional optimization problems, an ADPCCSO algorithm is proposed. First, to balance the breadth and depth search abilities of the basic CSO algorithm, an s-shaped function is utilized to adaptively adjust the value of parameter *G*. Then, in order to improve the solution accuracy of the algorithm, inspired by the literature [32], an improvement factor is used to dynamically adjust the foraging behaviors of chickens. At the same time, when the role-update condition is met, the chicks are arranged to search for food near the global optimal value, which can enhance the depth optimization ability of the algorithm. Finally, in view of the fact that the AFSA has unique behavior-pattern characteristics, which can make the algorithm quickly jump out of the local optimal solution in solving the high-dimensional optimization problems, it is integrated into the CSO algorithm to construct a dual-population collaborative optimization strategy based on chicken swarms and artificial fish swarms to enhance the global search ability, so as to achieve rapid optimization in the algorithm.

### 3.1. The Improvement Strategy for Parameter G

In the basic CSO algorithm, the parameter *G* determines how often the hierarchical relationship and role assignment of the chicken swarm are updated. The setting of an appropriate parameter *G* plays a crucial role in balancing the breadth and depth search abilities of the algorithm. Too large a value of *G* means that the information interaction between individuals is slow, which is not conducive to improving the breadth search ability of the algorithm. Too small a value of *G* will make the information interaction between individuals too frequent, which is not beneficial to enhancing the depth-optimization ability of the algorithm. Considering that the value of parameter *G* is a constant in the basic CSO algorithm, it is not conducive to balancing the search abilities between breadth and depth. We use Equation (7) to adaptively adjust the value of the parameter *G*; that is, in the early stage of the algorithm iteration, let *G* take a smaller value to enhance the breadth optimization ability of the algorithm; in the late stage of iteration of the algorithm, let *G* take a larger value to enhance the depth-optimization ability of the algorithm.
(7)G=round (40+60/(1+exp(15−0.5t)))
where *t* represents the current number of iterations and *round* () is a rounding function that can round an element to the nearest integer.

### 3.2. The Improvement Strategy for Foraging Behaviors

To improve the solution accuracy and depth-optimization ability of the algorithm, we construct an improvement strategy for foraging behaviors in this section; that is, an improvement factor is used in updating formulas of chickens. At the same time, in an effort to improve the depth optimization ability of CSO algorithm, the chicks’ foraging behavior near the optimal value is also added.

#### 3.2.1. Improvement Factor

To enhance the optimization ability of the algorithm, a learning factor was integrated into the foraging formula of roosters in Reference [32], which can be shown as follows:(8)$$a(t)=t\times \left(\log(\omega_{\max})-\log(\omega_{\min})\right)/M-\log(\omega_{\max})$$
(9)ω(t)=exp(−a(t))
where *M* is the maximum number of iterations and $\omega_{\max}$ and $\omega_{\min}$ are the maximum and minimum values of the learning factor, whose values are 0.9 and 0.4, respectively.

The method in Reference [32] improved the optimization ability of the algorithm to a certain degree, but it only modified the position update formula of roosters, which is not conducive to further optimization of the algorithm. Therefore, we slightly modified the learning factor in Reference [32] and named it the improvement factor; that is, through trial and error, we set the maximum and minimum values of the improvement factor to be 0.7 and 0.1, respectively, and then used them in the foraging formulas of roosters, hens, and chicks. The experimental results have demonstrated that the solution accuracy and convergence performance are significantly improved. The modified foraging formulas for roosters, hens, and chicks are shown in Equations (10)–(12):(10)$$X_{i,j}\left(t+1\right)=\omega (t)\times X_{i,j}\left(t\right)\times (1+Randn\left(0,\sigma^{2}\right))$$
(11)$$X_{i,j}\left(t+1\right)=\omega (t)\times X_{i,j}\left(t\right)+c_{1}\times rand()\times \left(X_{r_{1},j}\left(t\right)-X_{i,j}\left(t\right)\right)+c_{2}\times rand()\times \left(X_{best,j}(t)-X_{i,j}\left(t\right)\right)$$
(12)$$X_{i,j}\left(t+1\right)=\omega (t)\times X_{i,j}\left(t\right)+FL\times \left(X_{m,j}\left(t\right)-X_{i,j}\left(t\right)\right)+FL\times \left(X_{best,j}(t)-X_{i,j}\left(t\right)\right)$$

#### 3.2.2. Chicks’ Foraging Behavior near the Optimal Value

To enhance the depth optimization ability of the CSO algorithm, when the role update condition is met, chicks are allowed to search for food directly near the current optimal value. The corresponding formula is as follows:(13)$$X_{i,j}\left(t+1\right)=lb+\left(ub-lb\right)\times rand()$$
(14)$$lb=X_{best,j}(t)-\left|X_{best,j}(t)\right|\times rand\left(\right)$$
(15)$$ub=X_{best,j}(t)+\left|X_{best,j}(t)\right|\times rand\left(\right)$$
where $X_{best,j}(t)$ is the global optimal individual position at iteration *t*. *lb* and *ub* are the upper and lower bounds of an interval set near the current optimal value.

### 3.3. The Dual-Population Collaborative Optimization Strategy

To speed up the step of the algorithm jumping out of the local extrema, so as to quickly converge to the global optimal value, in view of the good robustness and global search ability of AFSA, the AFSA is introduced to construct a dual-population collaborative optimization strategy based on the chicken swarm and artificial fish swarm. With this strategy, the excellent individuals and several random individuals between the two populations are exchanged to break the equilibrium state within the population, so that the algorithm jumps out of the local extrema. The flow chart of the dual-population collaborative optimization strategy are shown in Figure 1.

The main steps are as follows:(1)Population initialization. Randomly generate two initial populations with a population size of *N*: the chicken swarm and the artificial fish swarm.(2)Chicken swarm optimization. Calculate the fitness function values of the entire chicken swarm and record the optimal value.
(a)Update the position of chickens.(b)Update the optimal value of the current chicken swarm.(3)Artificial fish swarm optimization. Calculate the fitness function values of the entire artificial fish swarm and record the optimal value.
(i)Update the positions of artificial fish swarm.

Update the positions of the artificial fish swarm; that is, by simulating fish behaviors of preying, swarming, and following, compare the fitness function values to find out the best behavior and execute this behavior. Their corresponding formulas are as follows.

The preying behavior:(16)$$X_{i|next}=X_{i}+rand\left(\right)\times Step\times \frac{X_{j}-X_{i}}{\left\|X_{j}-X_{i}\right\|}$$
(17)$$X_{j}=X_{i}+rand\left(\right)\times Visual$$
where *X_i_* is the position of the *i*th artificial fish. *Step* and *Visual* represent the step length and visual field of an artificial fish, respectively.

The swarming behavior:(18)$$X_{i|next}=X_{i}+rand\left(\right)\times Step\times \frac{X_{c}-X_{i}}{\left\|X_{c}-X_{i}\right\|}$$
(19)$$X_{c}=\frac{\sum_{ci=1}^{n_{f}}X_{ci}}{n_{f}}$$
where *n_f_* represents the number of partners within the visual field of the artificial fish. *X_c_* is the center position.

The following behavior:(20)$$X_{i|next}=X_{i}+rand\left(\right)\times Step\times \frac{X_{\max}-X_{i}}{\left\|X_{\max}-X_{i}\right\|}$$
where *X*_max_ is the position of an artificial fish with the optimal food concentration that can be found within the current artificial fish’s visual field.


(ii)Update the optimal value of the current artificial fish swarm.


(4)Interaction. To realize information interaction and thus break the equilibrium state within the population, first, select the optimal individuals in the chicken swarm and artificial fish swarm for exchange, and then select the remaining *Num* (*Num* < *N*) individuals randomly generated in the two populations for exchange.(5)Repeat steps (2)–(4) until the specified maximum number of iterations is reached and the optimal value is output.

### 3.4. The Design and Implementation of the ADPCCSO Algorithm

To address the premature convergence issue encountered by the basic CSO algorithm in solving high-dimensional optimization problems, the ADPCCSO algorithm is proposed. Firstly, the algorithm adjusts the parameter *G* adaptively and dynamically to balance the algorithm’s breadth and depth search ability. Then, the solution accuracy and depth-optimization ability of the algorithm are enhanced by using the improvement strategy for foraging behaviors described in Section 3.2. Finally, the dual-population collaborative optimization strategy is introduced to accelerate the step of the algorithm jumping out of the local extrema. The specific process is as follows:(1)Parameter initialization. The numbers of roosters, hens, and chicks are 0.2 × *N*, 0.6 × *N*, and *N* − 0.2 × *N −* 0.6 × *N*, respectively.(2)Population initialization. Initialize the two populations according to the method described in Section 3.3.(3)Chicken swarm optimization. Calculate the fitness function values of chickens and record the optimal value of the current population.(4)Conditional judgment. If *t* = 1, go to step (c); otherwise, execute step (a).
(a)Judgment of the information interaction condition in the chicken swarm. If *t*%*G* = 1, execute step (b); otherwise, go to step (d).(b)Chicks’ foraging behavior near the optimal value. Chicks search for food according to Equations (13)–(15) in Section 3.2.2.(c)Information interaction. In light of the current fitness function values of the entire chicken swarm, the dominance relationship and hierarchical relationship of the whole population are updated to achieve information interaction.(d)Foraging behavior. The chickens with different roles search for food according to Equations (10)–(12).(e)Modification of the optimal value in the chicken swarm: after each iteration, the optimal value of the whole chicken swarm is updated.(5)Artificial fish swarm optimization. Calculate the fitness function values of the artificial fish swarm and record the optimal value of the current population.
(i)In the artificial fish swarm, behaviors of swarming, following, preying, and random movement are executed to find the optimal food.(ii)Update the optimal value of the whole artificial fish swarm.(6)Exchange. This includes the exchange of the optimal individuals and the exchange of several other individuals in the two populations.(7)Judgment of ending condition for the algorithm. If the specified maximum number of iterations is reached, the optimal value will be output, and the program will be terminated. Otherwise, go to step (3).

### 3.5. The Time Complexity Analysis of the ADPCCSO Algorithm

In the standard CSO algorithm, if the population size of the chicken swarm is assumed to be *N*, then the dimension of the solution space is *d*, the iteration number of the entire algorithm is *M*, and the hierarchical relationship of the chicken swarm is updated every *G* iterations. The numbers of roosters, hens, and chickens in the chicken swarm e *N_r_*, *N_h_*, and *N_c_*, respectively; that is, *N_r_* + *N_h_* + *N_c_* = *N*. The calculation time of the fitness function value of each chicken is *t_f_*. Therefore, the time complexity of the CSO algorithm consists of two stages, namely, the initialization stage and the iteration stage [30,32].

In the initialization stage (including parameter initialization and population initialization), assume that the setting time of parameters is *t*_1_, the actual time required to generate a random number is *t*_2_, and the sorting time of the fitness function values is *t*_3_. Then, the time complexity of the initial stage is T_1_ = *t*_1_ + *N* × *d* × *t*_2_+ *t*_3_ + *N × t_f_* = O(*N* × *d + N* × *t_f_*).

In the iteration stage, let the time for each rooster, hen, and chick to update its position on each dimension be *t_r_*, *t_h_*, and *t_c_*, respectively. The time it takes to compare the fitness function values between two individuals is *t*_4_, and the time it takes for the chickens to interact with information is *t*_5_. Therefore, the time complexity of this stage is as follows.
$$T_{2}=M\times \;d\times \;N_{r}\times t_{r}+\;M\times \;d\times \;N_{h}\times t_{h}+\;M\times \;d\times \;N_{c}\times t_{c}+\;N\times \;M\times \;t_{f}+\;M\times \;N\times t_{4}\;+\frac{M}{G}\;\times t_{5}=M\times \;d\times \;(N_{r}\times t_{r}+\;N_{h}\times \;t_{h}+\;N_{c}\times t_{c})+\;N\times \;M\times \;(t_{f}+\;t_{4})+\frac{M}{G}\times \;t_{5}\;=\;O(N\times \;M\times \;d+\;N\times \;M\times \;t_{f}).$$

Therefore, the time complexity of the standard CSO algorithm is as follows.
T′ = T_1_ + T_2_ = O(*N* × *d* + *N* × *t_f_*) + O(*N* × *M* × *d* + *N* × *M* × *t_f_*) = O(*N* × *M* × *d* + *N* × *M* × *t_f_*).

On the basis of the standard CSO algorithm, the ADPCCSO algorithm adds the improvement factor in the position update formula of the chicken swarm, the foraging behavior of chicks near the optimal value, and the optimization strategy of the artificial fish swarm. It is assumed that the population size of the artificial fish swarm is *N*, and the tentative number when performing foraging behavior is *try_number*. In the swarming and following behaviors, it is necessary to count *friend_number* times when calculating the values of *n_f_* and *X_max_*. The time to calculate the improvement factor is *t*_6_, and the time it takes to perform the foraging, swarming, and following behaviors are *t*_7_, *t*_8,_ and *t*_9_, respectively.

Therefore, the time complexity of adding the improvement factor in the position updating formula is T_3_ = *M* × *N* × *t*_6_ = O(*M* × *N*). The time complexity of the chicks’ foraging behavior near the optimal value is T_4_ = $\frac{M}{G}$ × *d* × *N_c_* × *t_c_* = O($\frac{M}{G}$ × *d* × *N_c_*).

The time complexity of the artificial fish swarm optimization strategy is mainly composed of three parts: foraging behavior, swarming behavior, and following behavior. Its time complexity is as follows [33].
T_5_ = *M* × *N* × *try_number* × *t*_7_ × *d* + *M* × *N* × *t*_8_ × *Friend_number* × *d* + *M* × *N* × *Friend_number* × *t*_9_ × *d* = O(*M* × *N* × *try_number* × *d*) + O(*M* × *N* × *Friend_number* × *d*) + O(*M* × *N* × *Friend_number* × *d*) = O(*M* × *N* × *d*).

Therefore, the time complexity of the ADPCCSO algorithm is as follows.
$$T\;=\;T^{\prime }\;+\;T_{3}\;+\;T_{4}\;+\;T_{5}\;=\;O(N\;\times \;M\times \;d\;+N\times \;M\times \;t_{f})\;+\;O(M\;\times \;N)\;+\;O(\frac{M}{G}\;\times \;d\;\times N_{c})\;=\;O(N\;\times \;M\times \;d\;+N\times \;M\times \;t_{f})$$

It can be seen that the time complexity of the ADPCCSO and standard CSO algorithms is still in the same order of magnitude.

## 4. Simulation Experiment and Analysis

### 4.1. The Experimental Setup

In this study, our experiments were conducted on a desktop computer with an Intel^®^ Pentium^®^ CPU G4500 @ 3.5 GHz processor, 12 GB RAM, a Windows 7 operating system, and the programming environment of MATLABR2016a.

To verify the performance of he ADPCCSO algorithm in solving high-dimensional complex optimization problems, we selected 17 standard high-dimensional test functions in Reference [28] for experimental comparison, which are listed in Table 1. (Because the functions *f*_18_~*f*_21_ in Reference [28] are fixed low-dimensional functions, we only selected the functions *f*_1_~*f*_17_ for experimental comparison.) Here, the functions *f*_1_~*f*_12_ are unimodal functions. Because it is difficult to obtain the global optimal solution, they are often used to test the solution accuracy of the algorithms. The functions *f*_13_~*f*_17_ are multimodal functions, which are often used to verify the global optimization ability of the algorithms.

To fairly compare the performance of various algorithms, we need to make all algorithms have the same number of function evaluations (FEs). In our paper, FEs = the population size × the maximum number of iterations, and considering that the population size and the maximum number of iterations of GCSO [27] and DMCSO [29] are both 100 and 1000, in the experiment, we also set these two parameters for the remaining algorithms to 100 and 1000, respectively. The experimental data in this paper are obtained by independently running all algorithms on each function for 30 times. Other parameter settings are shown in Table 2.

In Table 2, c_1_ and c_2_ are two learning factors and $\omega_{\min}$ and $\omega_{\max}$ are the upper and lower bounds of the inertial weight. *hPercent* and *rPercent* are the proportion of hens and roosters in the entire chickens, respectively. *N_c_*, *N_re_*, and *N_ed_* represent the numbers of chemotactic, reproduction, and elimination-dispersal operations, respectively. *Visual*, *Step*, and *try_number* represent the vision field, step length, and maximum tentative number of the artificial fish swarm, respectively. *Limit* is a control parameter for bees to abandon their food sources. *P_c_* and *P_m_* are crossovers and variation operators.

In Table 2, the parameters of AFSA are set after trial and error on the basis of the literature [31]. The parameter of ABC is set according to the study [34] where ABC has been proposed. The parameters of PSO, CSO, ASCSO-S [28], GCSO [27], and DMCSO [29] are set according to their corresponding references (namely the studies [27,28,29]), respectively.

### 4.2. The Effectiveness Test of Two Improvement Strategies

To verify the effectiveness of the two improved strategies proposed in Section 3.1 and Section 3.3, we have compared the ACSO, DCCSO, and CSO algorithms on 17 test functions in terms of the solution accuracy and convergence performance. Here, the ACSO algorithm is an adaptive CSO algorithm, that is, we only use Equation (7) to make adaptive dynamic adjustment to the parameter *G* in the CSO algorithm. The DCCSO algorithm refers to the fact that only the dual-population collaborative optimization strategy mentioned in Section 3.3 is used in CSO algorithm.

The experimental results of the above three algorithms on 17 test functions are listed in Table 3, where the optimal results are marked in bold. In Table 3, “Dim” is the dimension of the problem to be solved, “Mean” is the mean value, and “Std” is the standard deviation. “↑”, “↓”, and “=”, respectively, signify that the operation results obtained by the ACSO and DCCSO algorithms are superior to, inferior to, and equal to those obtained by the basic CSO algorithm.

It can be seen from Table 3 that the optimization results of the ACSO and DCCSO algorithms on almost all benchmark test functions are far superior to those of the CSO algorithm (on only function *f*_2_, the optimization results of DCCSO algorithm are slightly inferior to those of CSO algorithm); in particular, the experimental data on functions *f*_10_ and *f*_11_ reached the theoretical optimal values. This shows the effectiveness of the two improvement strategies proposed in Section 3.1 and Section 3.3 in terms of solution accuracy.

To verify the effectiveness of ACSO and DCCSO algorithms compared with the CSO algorithm in terms of the aspect of convergence performance, the convergence curves of the above three algorithms on some functions are shown in Figure 2. For simplicity, we only list the convergence curves of the aforementioned algorithms on functions *f*_1_, *f*_9_, *f*_13_, and *f*_16_, where functions *f*_1_ and *f*_9_ are unimodal functions and functions *f*_13_ and *f*_16_ are multimodal functions. In addition, in order to make the convergence curves clearer, we take the logarithmic processing for the average fitness values.

As can be seen from Figure 2, the convergence performance of both ACSO and DCCSO algorithms is significantly superior to that of the CSO algorithm, which proves the effectiveness of the two improvement strategies proposed in this paper in terms of convergence performance.

### 4.3. The Effectiveness Test of Improvement Strategy for Foraging Behaviors

To test the effectiveness of the improvement strategy proposed in Section 3.2, the learning-factor-based foraging behavior improvement strategy in the literature [32] is used for experimental comparison. At the same time, with the purpose of conducting experimental comparison more objectively and fairly, we let the ADPCCSO algorithm use the above-mentioned improvement strategies on 17 test functions to verify the performance of the improvement strategy in Section 3.3. The experimental results are listed in Table 4, where the ADPCCSO [32] indicates that the improvement strategy for foraging behavior in the literature [32] is used in the ADPCCSO. In addition, the number of optimal results calculated by each algorithm based on the mean value is also shown in Table 4.

As can be seen from Table 4, the ADPCCSO [32] only obtained optimal values on 5 functions, while the ADPCCSO algorithm obtained optimal values on 16 functions and the theoretical optimal values were obtained on 13 functions. Only on function *f*_5_ were the results of ADPCCSO algorithm slightly inferior to those of the ADPCCSO [32]. This shows the effectiveness of the improvement strategy proposed in Section 3.2 in terms of solution accuracy.

To test the effectiveness of the improvement strategy proposed in Section 3.3 in terms of convergence performance, the convergence curves of the above two algorithms are also listed in this section. For simplicity, only their convergence curves on functions *f*_9_ and *f*_15_ are given, which are shown in Figure 3. It is worth noting that in order to make the convergence curves look more intuitive and clearer, we also logarithm the average fitness values in this section.

It is obvious from Figure 3 that the convergence performance of the ADPCCSO algorithm is better than that of ADPCCSO [32] as a whole. Especially on function *f*_15_, the ADPCCSO algorithm has more obvious advantages in convergence performance, and it began to converge stably around the 18th generation.

### 4.4. Performance Comparison of Several Swarm Intelligence Algorithms

To test the advantages of the ADPCCSO algorithm proposed in this paper over other algorithms in solving high-dimensional optimization problems, in this section, it is compared with five other algorithms, namely ASCSO-S [28], ABC, AFSA, CSO, and PSO. Their best values, worst values, mean values, and standard deviations obtained on the 17 benchmark standard test functions are shown in Table 5, Table 6 and Table 7, where the best values are shown in bold. In addition, we also count the number of optimal values obtained by each algorithm based on the mean value, which are shown in Table 5, Table 6 and Table 7.

It is not difficult to see from Table 5, Table 6 and Table 7 that the ADPCCSO and ASCSO-S algorithms are far superior to the other four swarm intelligence algorithms in terms of solution accuracy and stability. Among them, the ADPCCSO algorithm has the best performance: in particular, when Dim = 500, it obtained the optimal values in all 17 functions, and the number of optimal results calculated by the ASCSO-S algorithm is 14. Additionally, on function *f*_5_, the operation results of the ADPCCSO algorithm at Dim = 100 and Dim = 500 are far better than those at Dim = 30, which also shows to a certain extent that the ADPCCSO algorithm is more suitable for handling higher-dimensional complex optimization problems.

As can be seen from Table 5, although the ABC algorithm obtained the optimal values in three functions, its optimization ability worsens as the dimension of the problem increases. On the contrary, AFSA shows a higher optimization ability (when Dim = 500, its optimization ability on 11 functions is much better than that of the ABC algorithm), which is one of the reasons why we constructed a dual-population collaborative optimization strategy based on a chicken swarm and an artificial fish swarm to solve high-dimensional optimization problems. It is noteworthy that the operation results of the PSO algorithm on function *f*_8_ are not given in Table 7. This is because when Dim = 500, its fitness function values often exceed the maximum positive value that the computer can represent, resulting in the algorithm being unable to obtain suitable operation results. This also shows that the PSO algorithm is not suitable for handling higher-dimensional complex optimization problems.

Below, we summarize why the solution accuracy of ADPCCSO and ASCSO-S algorithms is better than that of the other four algorithms. This may be due to the fact that both algorithms introduce an improvement factor (which is called an inertial weight) into the position update formula of the chicken swarm. The reason why the performance of the former in terms of solution accuracy is better than that of the latter may be because the former uses an improvement strategy for foraging behaviors, which not only improves the depth optimization ability of the algorithm but also improves its solution accuracy.

To verify the superiority of the ADPCCSO algorithm over other algorithms in terms of convergence performance, this paper presents the convergence curves of the above six algorithms on all 17 test functions with Dim = 100, which are shown in Figure 4. In Figure 4, the average fitness values of all ordinates are also logarithmic. In addition, in order to further present a clearer convergence effect, we have locally enlarged some convergence curves, which is why there are subgraphs in some convergence curves.

As can be seen from Figure 4, the ADPCCSO algorithm has the best convergence performance on 16 functions, but on only function *f*_4_, its convergence is slightly inferior to that of the ABC algorithm. ASCSO-S ranks second in terms of convergence performance, and AFSA and CSO are tied for third place. (This is another reason why we construct a dual-population collaborative optimization strategy based on the chicken swarm and artificial fish swarm).

Below, we summarize why the convergence performance of the ADPCCSO and ASCSO-S algorithms is better than that of the other four algorithms as a whole. This may be because both algorithms use adaptive dynamic adjustment strategies. The convergence performance of the former is superior to that of the latter, which may be due to the use of the dual-population collaborative optimization strategy in the ADPCCSO algorithm, which improves the convergence performance of the algorithm. In addition, by carefully observing Figure 4, it is not difficult to find that on functions *f*_1_–*f*_3_, *f*_6_–*f*_8_, *f*_10_–*f*_14,_ and *f*_16_–*f*_17_, it seems that the convergence curves of ADPCCSO and ASCSO-S algorithms in the late iteration stage are not fully presented. This is because both algorithms have found the theoretical optimal value of 0 in these functions, and 0 has no logarithm.

### 4.5. Friedman Test of Algorithms

The Friedman test, or Friedman’s method for randomized blocks, is a non-parametric test method that does not require the sample to obey a normal distribution, and it only uses rank to judge whether there are significant differences in multiple population distributions. This method was proposed by Friedman in 1973. Because of its simple operation and no strict requirements for data, it is often used to test the performance of algorithms [28,35].

To further test the performance of the ADPCCSO algorithm proposed in this paper, in this section, the Friedman test is utilized to compare the performance of the above six algorithms from a statistical perspective. For the minimum optimization problem, the smaller the average ranking of the algorithm is, the better the performance of the algorithm is. In this section, the SPSS software is used to calculate the average ranking values of all algorithms. The statistical results are shown in Table 8. It is obvious from Table 8 that the ADPCCSO algorithm has the lowest average ranking of 1.5 and therefore has the best performance.

### 4.6. Performance Comparison of Several Improved CSO Algorithms

To further verify the performance of ADPCCSO algorithm proposed in this paper, two improved CSO algorithms mentioned in the literature [27,29], namely GCSO [27] and DMCSO [29], have also been used to compare with the ADPCCSO algorithm. The experimental results are shown in Table 9. The experimental data of both algorithms are from the corresponding references. It is worth noting that the population size of the above three algorithms is 100, and the maximum number of iterations is 1000, which also facilitates a more fair and reasonable experimental comparison. Other parameter settings are shown in Table 2.

In Table 9, GCSO [27] counts the operation results of 6 functions out of 17 test functions but only obtained the optimal values on the standard deviation of function *f*_4_ and the best values of functions *f*_13_ and *f*_14_. DMCSO [29] counted the operation results of 12 functions out of 17 test functions and only obtained the optimal values on function *f*_4_. However, the operation results of the ADPCCSO algorithm are better than those of the above two algorithms overall. On only function *f*_4_, the operation results of the ADPCCSO algorithm are worse than those of DMCSO [29]. This shows the advantages of the ADPCCSO algorithm.

### 4.7. Performance Test of ADPCCSO Algorithm for Solving Higher-Dimensional Problems

To further verify the performance of the ADPCCSO algorithm in solving higher-dimensional optimization problems, the relevant experiments for the proposed algorithm on 17 benchmark test functions with Dim = 1000 are also presented in this section. The corresponding experimental results are shown in Table 10.

As can be seen from Table 10, even when the dimension of the optimization problem is adjusted to 1000, the proposed algorithm can still achieve satisfactory optimization accuracy on most test functions; only on functions *f*_4_, *f*_5_, and *f*_9_ do the experimental data fluctuate slightly. This indicates that when the dimension increases, the proposed algorithm will not be greatly affected, which fully demonstrates that the ADPCCSO algorithm still has a competitive advantage in dealing with higher-dimensional optimization problems.

### 4.8. Parameter Estimation Problem of Richards Model

To verify the performance of ADPCCSO algorithm in solving practical problems, it is applied to the parameter estimation problem of the Richards model in this section. The Richards model is a growth curve model with four unknown parameters, which can adequately simulate the whole process of biological growth. Its mathematical formula is as follows [28,36,37]:(21)$$y(t)=\alpha {\left(1-e^{\beta -\gamma t}\right)}^{-\frac{1}{\delta }}$$
where y(t) stands for the growth amount at time *t*, and α,β,γ,δ are four unknown parameters.

The core problem of the ADPCCSO algorithm for parameter estimation of the Richards model is the design of the fitness function. In this paper, the fitness function design method mentioned in the sties [28,36] is adopted; that is, the sum of squares of the difference between the observed and predicted values is used as the fitness function. The mathematical formula is as follows:(22)$$fit(\alpha ,\beta ,\gamma ,\delta )=\sum_{i=1}^{n}{\left(y_{i}-\alpha {\left(1+e^{\beta -\gamma t_{i}}\right)}^{-\frac{1}{\delta }}\right)}^{2}$$
where *y_i_* is the actual growth amount observed at time *i*. In this section, the actual growth concentrations of glutamate listed in the studies [28,36] are used as the observation values, which are shown in Table 11. The optimal solutions obtained by different algorithms through 30 independent runs are listed in Table 12, where the experimental data of ASCSO-S [28] and VS-FOA [36] come from the corresponding references. The data in Table 13 are the growth concentration of glutamate calculated by using the data in Table 12 in Equation (21). In Table 13, “fit” represents the fitness function value.

To evaluate the effect of VS-FOA [36], ASCSO-S [28], and ADPCCSO in the parameter estimation of Richards model, we select the root mean square error, mean absolute error, and coefficient of determination as evaluation indexes to evaluate the performance of the above three algorithms. The formulas are as follows:

(1) The root mean square error:(23)$$\mathrm{RMSE}=\sqrt{\frac{\sum_{i=1}^{n}(y_{i}-{\hat{y}}_{i}{)}^{2}}{n}}$$
where *y_i_* is the actual value observed and ${\hat{y}}_{i}$ is the predicted value at time *i*. *n* is the number of actual values observed. The root mean square error is used to measure the deviation between the predicted values and the observed values. The smaller its value is, the better the predicted value is.

(2) The mean absolute error:(24)$$\mathrm{MAE}=\frac{1}{n}\sum_{i=1}^{n}\left|(y_{i}-{\hat{y}}_{i}\right)|$$

The mean absolute error is the mean value of the absolute error. It reflects the actual situation of the error of the predicted value better. The smaller its value is, the more precise the predicted value is.

(3) The coefficient of determination:(25)$$R^{2}=1-\frac{\sum_{i=1}^{n}({\hat{y}}_{i}-y_{i}{)}^{2}}{\sum_{i=1}^{n}(y_{i}-\bar{y}{)}^{2}}$$
where $\bar{y}$ is the mean value of the actual values observed. The coefficient of determination is generally used to evaluate the conformity between the predicted and actual values. The closer its value is to 1, the better the prediction effect.

The comparison results of the above three algorithms in the three evaluation indexes are shown in Table 14, where the optimal values are marked in bold.

As can be seen from Table 14, the ADPCCSO algorithm has optimal values in both indexes. Although the ADPCCSO algorithm is slightly inferior to the other two algorithms in terms of the mean absolute error, its fitness function value is indeed the best of the three algorithms, which can be seen from Table 13. This preliminarily shows that the ADPCCSO algorithm can solve the parameter estimation problem of the Richards model.

## 5. Conclusions

In view of the precocious convergence problem that the basic CSO algorithm is prone to when solving high-dimensional complex optimization problems, an ADPCCSO algorithm is proposed in this paper. The algorithm first uses an adaptive dynamic adjustment method to give the value of parameter *G*, so as to balance the algorithm’s depth and breadth search ability. Additionally, then, the solution accuracy and depth optimization ability of the algorithm are improved by using a foraging-behavior-improvement strategy. Finally, a dual-population collaborative optimization strategy is constructed to improve the algorithm’s global search ability. The experimental results preliminarily show that the proposed algorithm has obvious advantages over other comparison algorithms in terms of solution accuracy and convergence performance. This provides new ideas for the study of high-dimensional optimization problems.

However, although the experimental results of the proposed algorithm on most given benchmark test functions have achieved obvious advantages over the comparison algorithms, there is still a gap between the actual optimal solutions obtained on several functions and their theoretical optimal solutions. Therefore, understanding how to improve the performance of the algorithm to better solve more complex large-scale optimization problems still needs further research. Moreover, in future research work, it is also a good choice to apply this algorithm to other fields, such as the constrained optimization problem, the multi-objective optimization problem, and the vehicle-routing problem.

## Figures and Tables

**Figure 1 biomimetics-08-00210-f001:**
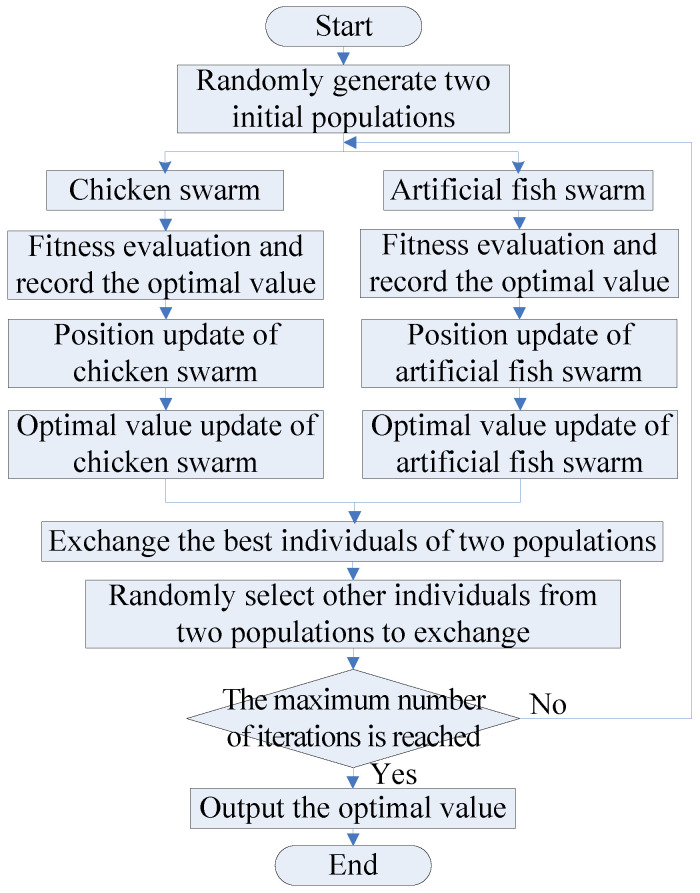
The flow chart of the dual-population collaborative optimization strategy.

**Figure 2 biomimetics-08-00210-f002:**
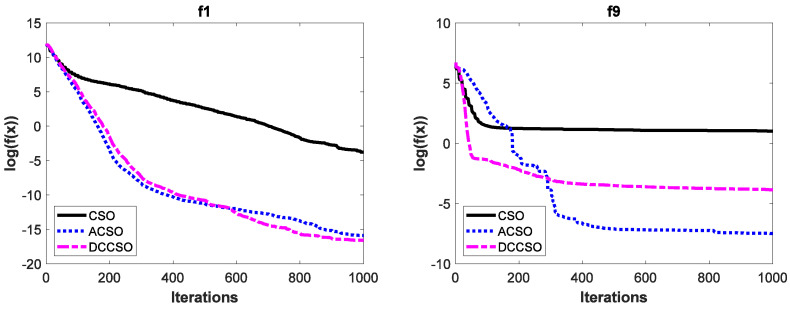
The convergence curves of three algorithms on functions *f*_1_, *f*_9_, *f*_13_ and *f*_16_.

**Figure 3 biomimetics-08-00210-f003:**
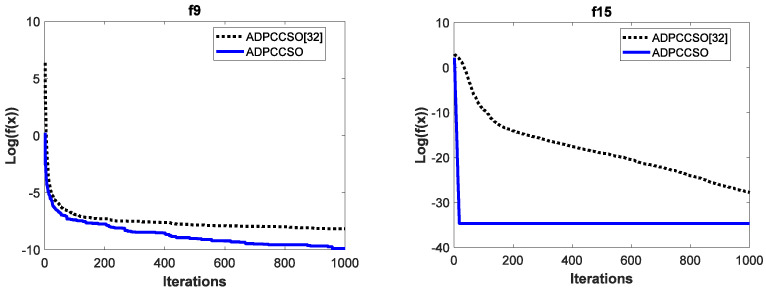
The convergence curves of two algorithms on functions *f*_9_ and *f*_15_.

**Figure 4 biomimetics-08-00210-f004:**
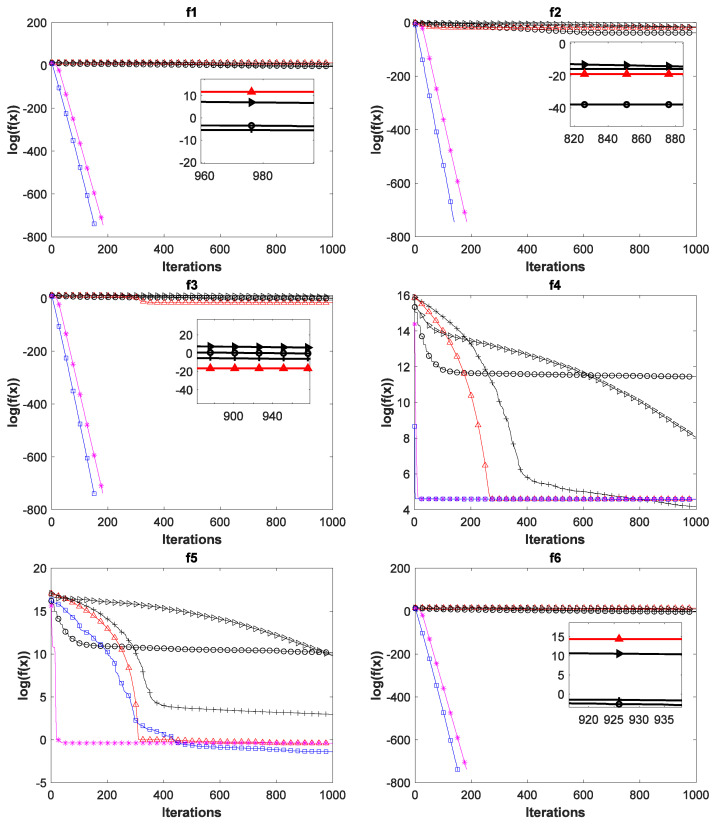
Convergence curves of six swarm intelligence algorithms on 17 functions.

**Table 1 biomimetics-08-00210-t001:** The description of the test functions.

Type	Functions	Names	Search Ranges
Unimodal Functions	$f_{1}=\sum_{i=1}^{n}\chi_{i}^{2}$	Sphere	[−100,100]
	$f_{2}=\sum_{i=1}^{n}{\left|\chi_{i}\right|}^{i+1}$	Sum of different powers	[−1,1]
	$f_{3}=\sum_{i=1}^{n}i\chi_{i}^{2}$	Sum squares	[−10,10]
	$f_{4}=\sum_{i=1}^{n-1}\left[100{\left(\chi_{i+1}-\chi_{i}^{2}\right)}^{2}+{\left(\chi_{i}-1\right)}^{2}\right]$	Rosenbrock	[−5,10]
	$f_{5}={\left(\chi_{1}-1\right)}^{2}+{\sum_{i=2}^{n}i\left(2\chi_{i}^{2}-\chi_{i-1}\right)}^{2}$	Dixon-price	[−10,10]
	$f_{6}=\sum_{i=1}^{n}\sum_{j=1}^{i}\chi_{j}^{2}$	Rotated hyper-ellipsoid	[−65.536,65.536]
	$f_{7}=\max\left|\chi_{i}\right|$	Schwefels P 2.21	[−100,100]
	$f_{8}=\sum_{i=1}^{n}\left|\chi_{i}\right|+\prod_{i=1}^{n}\left|\chi_{i}\right|$	Schwefels P 2.22	[−10,10]
	$f_{9}=\sum_{i=1}^{n}i\chi_{i}^{4}+rand\left[0,1\right)$	Quartic	[−1.28,1.28]
	$f_{10}={\sum_{i=1}^{n}\left(\left[\chi_{i}+0.5\right]\right)}^{2}$	Step	[−100,100]
	$f_{11}=10^{6}\chi_{1}^{2}\sum_{i=1}^{n}\chi_{i}^{2}$	Discus	[−100,100]
	$f_{12}=\sum_{i=1}^{n}\chi_{i}^{2}+{\left(\sum_{i=1}^{n}0.5i\chi_{i}\right)}^{2}+{\left(\sum_{i=1}^{n}0.5i\chi_{i}\right)}^{4}$	Zakharov	[−5,10]
Multimodal Functions	$f_{13}=\frac{1}{4000}\sum_{i=1}^{n}\chi_{i}^{2}-\prod_{i=1}^{n}\cos\left(\frac{\chi_{i}}{\sqrt{i}}\right)+1$	Griewank	[−600,600]
	$f_{14}=\sum_{i=1}^{n}\left[\chi_{i}^{2}-10\cos\left(2\pi \chi_{i}\right)+10\right]$	Rastrigin	[−5.12,5.12]
	$f_{15}=-20\exp\left(-0.2\sqrt{\frac{\sum_{i=1}^{n}\chi_{i}^{2}}{n}}\right)-\exp\left(\frac{1}{n}\sum_{i=1}^{n}\cos\left(2\pi \chi_{i}\right)\right)+20+e$	Ackley	[−32,32]
	$f_{16}=\sum_{i=1}^{n/4}\left[{\left(\chi_{4i-3}+10\chi_{4i-2}\right)}^{2}+5{\left(\chi_{4i-1}-\chi_{4i}\right)}^{2}+{\left(\chi_{4i-2}-2\chi_{4i-1}\right)}^{4}+10{\left(\chi_{4i-3}+\chi_{4i}\right)}^{4}\right]$	Powell	[−4,5]
	$f_{17}=\sum_{i=1}^{n}\left|\chi_{i}\sin\left(\chi_{i}\right)+0.1\chi_{i}\right|$	Alpine	[−10,10]
Global minimum: 0			

**Table 2 biomimetics-08-00210-t002:** The parameter settings of all algorithms.

Algorithms	Parameter Settings
PSO	c_1_ = c_2_ = 2, $\omega_{\min}=0.4$, $\omega_{\max}=0.9$
CSO	*rPercent* = 0.2, *hPercent* = 0.6, *G* = 10
AFSA	*Visual* = 2.5, *Step* = 0.3, *try_number* = 5
ABC	*Limit* = 100,
GCSO [27]	*rPercent* = 0.2, *hPercent* = 0.6, *G* = 10, *P_c_* = 0.8, *P_m_* = 0.2
ASCSO-S [28]	*rPercent* = 0.4, *hPercent* = 0.6, *G* = 100,
DMCSO [29]	*rPercent* = 0.2, *hPercent* = 0.6, *G* = 10
ADPCCSO	*rPercent* = 0.2, *hPercent* = 0.6

**Table 3 biomimetics-08-00210-t003:** The experimental comparison of two improvement strategies with Dim = 100.

Functions	Results	ACSO	DCCSO	CSO
*f* _1_	Mean	1.2641 × 10^−7^↑	**6.1971 × 10^−8^** **↑**	0.0231
	Std	5.0764 × 10^−7^	**3.3534 × 10^−7^**	0.0724
*f* _2_	Mean	**2.0509 × 10^−33^** **↑**	4.9656 × 10^−12^↓	3.6821 × 10^−17^
	Std	**1.1232 × 10^−32^**	1.2131 × 10^−11^	2.0117 × 10^−16^
*f* _3_	Mean	**1.5007 × 10^−8^** **↑**	4.9472 × 10^−8^↑	0.5679
	Std	**3.5575 × 10^−8^**	1.7774 × 10^−7^	3.0256
*f* _4_	Mean	97.9332↑	**97.7092**↑	9.3677 × 10^4^
	Std	0.6317	**0.3845**	4.9125 × 10^4^
*f* _5_	Mean	0.6671↑	**0.2500**↑	2.6486 × 10^4^
	Std	**4.4377 × 10^−4^**	0.1451	1.5023 × 10^4^
*f* _6_	Mean	1.3338 × 10^−9^↑	**2.2178 × 10^−10^** **↑**	0.0374
	Std	6.4294 × 10^−9^	**5.8728 × 10^−10^**	0.1085
*f* _7_	Mean	26.3562↑	**22.9327**↑	27.6541
	Std	**2.8717**	4.3603	3.0137
*f* _8_	Mean	**8.3801 × 10^−28^** **↑**	6.1715 × 10^−27^↑	1.8973 × 10^−16^
	Std	**2.3832 × 10^−27^**	1.3900 × 10^−26^	2.8673 × 10^−16^
*f* _9_	Mean	**5.6485 × 10^−4^** **↑**	0.0213↑	2.7873
	Std	**0.0016**	0.0209	1.4489
*f* _10_	Mean	**0**↑	**0**↑	208.3000
	Std	**0**	**0**	608.1095
*f* _11_	Mean	**0=**	**0=**	**0**
	Std	**0**	**0**	**0**
*f* _12_	Mean	27.6333↑	**0.0036**↑	121.6425
	Std	8.7731	**0.0196**	31.6731
*f* _13_	Mean	1.7590 × 10^−6^↑	**5.1184 × 10^−9^** **↑**	0.0971
	Std	5.6806 × 10^−6^	**1.6819 × 10^−8^**	0.2474
*f* _14_	Mean	**2.3473 × 10^−11^** **↑**	1.6175 × 10^−10^↑	3.5819 × 10^−7^
	Std	**7.0250 × 10^−11^**	6.0763 × 10^−10^	1.3432 × 10^−6^
*f* _15_	Mean	**1.3016 × 10^−5^** **↑**	1.4443 × 10^−4^↑	4.2376
	Std	**3.1445 × 10^−5^**	6.7327 × 10^−4^	3.0389
*f* _16_	Mean	**4.5474 × 10^−5^** **↑**	1.8637 × 10^−4^↑	124.7637
	Std	**1.0932 × 10^−4^**	7.2074 × 10^−4^	80.3793
*f* _17_	Mean	**1.2160 × 10^−25^** **↑**	8.6956 × 10^−24^↑	0.1021
	Std	**4.4244 × 10^−25^**	3.9926 × 10^−23^	0.0795
↑	—	**16**	15	—
=	—	**1**	1	—
↓	—	**0**	1	—

**Table 4 biomimetics-08-00210-t004:** The experimental results of improvement strategy for foraging behaviors.

Functions	Dim	Results	ADPCCSO [32]	ADPCCSO
*f* _1_	100	Mean	4.5239 × 10^−29^	**0**
		Std	1.5994 × 10^−28^	**0**
*f* _2_	100	Mean	1.9065 × 10^−120^	**0**
		Std	1.0442 × 10^−119^	**0**
*f* _3_	100	Mean	1.4728 × 10^−29^	**0**
		Std	5.7490 × 10^−29^	**0**
*f* _4_	100	Mean	97.4457	**97.3542**
		Std	0.4394	**0.1748**
*f* _5_	100	Mean	**0.2080**	0.2483
		Std	**0.0360**	0.1220
*f* _6_	100	Mean	4.6786 × 10^−27^	**0**
		Std	2.4167 × 10^−26^	**0**
*f* _7_	100	Mean	4.5660 × 10^−9^	**0**
		Std	2.2296 × 10^−8^	**0**
*f* _8_	100	Mean	2.5173 × 10^−18^	**0**
		Std	1.2607 × 10^−17^	**0**
*f* _9_	100	Mean	2.8945 × 10^−4^	**5.0547 × 10^−5^**
		Std	2.2222 × 10^−4^	**4.0937 × 10^−5^**
*f* _10_	100	Mean	**0**	**0**
		Std	**0**	**0**
*f* _11_	100	Mean	**0**	**0**
		Std	**0**	**0**
*f* _12_	100	Mean	9.1771 × 10^−8^	**0**
		Std	4.5102 × 10^−7^	**0**
*f* _13_	100	Mean	**0**	**0**
		Std	**0**	**0**
*f* _14_	100	Mean	**0**	**0**
		Std	**0**	**0**
*f* _15_	100	Mean	8.3743 × 10^−13^	**8.8818 × 10^−16^**
		Std	2.1757 × 10^−12^	**0**
*f* _16_	100	Mean	2.1177 × 10^−13^	**0**
		Std	1.1599 × 10^−12^	**0**
*f* _17_	100	Mean	6.1813 × 10^−19^	**0**
		Std	2.9778 × 10^−18^	**0**
The number of optimal values	—	—	5	16

**Table 5 biomimetics-08-00210-t005:** The experimental results of several algorithms on the 17 test functions with Dim = 30.

Functions	Results	PSO	CSO	ABC	AFSA	ASCSO-S [28]	ADPCCSO
*f* _1_	Best	3.7892 × 10^−7^	1.9607 × 10^−57^	1.6335 × 10^−10^	5.6533 × 10^3^	**0**	**0**
	Worst	3.3136 × 10^−5^	3.5835 × 10^−51^	6.1183 × 10^−9^	1.1818 × 10^4^	**0**	**0**
	Mean	8.1646 × 10^−6^	2.0326 × 10^−52^	1.1637 × 10^−9^	8.6369 × 10^3^	**0**	**0**
	Std	7.3589 × 10^−6^	6.6738 × 10^−52^	1.2091 × 10^−9^	1.7458 × 10^3^	**0**	**0**
*f* _2_	Best	2.8786 × 10^−25^	3.4757 × 10^−229^	1.0079 × 10^−16^	1.7090 × 10^−14^	**0**	**0**
	Worst	1.1064 × 10^−19^	1.3980 × 10^−175^	3.4341 × 10^−12^	1.0720 × 10^−6^	**0**	**0**
	Mean	7.0544 × 10^−21^	4.6607 × 10^−177^	2.3088 × 10^−13^	8.7408 × 10^−8^	**0**	**0**
	Std	2.1066 × 10^−20^	**0**	6.5800 × 10^−13^	2.2756 × 10^−7^	**0**	**0**
*f* _3_	Best	2.2371 × 10^−8^	5.1238 × 10^−59^	5.6509 × 10^−12^	9.4436 × 10^−10^	**0**	**0**
	Worst	2.9967 × 10^−6^	6.0429 × 10^−50^	1.5907 × 10^−10^	2.5102 × 10^−5^	**0**	**0**
	Mean	6.2016 × 10^−7^	2.2297 × 10^−51^	4.7148 × 10^−11^	1.9605 × 10^−6^	**0**	**0**
	Std	6.9470 × 10^−7^	1.1016 × 10^−50^	3.8334 × 10^−11^	5.5333 × 10^−6^	**0**	**0**
*f* _4_	Best	16.2996	28.1179	**0.0172**	28.6682	28.0946	26.0670
	Worst	116.2702	28.8012	**2.0188**	28.6958	28.8361	26.4449
	Mean	40.4342	28.6045	**0.4208**	28.6815	28.5759	26.2860
	Std	28.1404	0.1780	0.4832	0.0064	0.1895	**0.1059**
*f* _5_	Best	0.1584	0.6667	**0.0020**	0.2365	0.6675	0.1264
	Worst	3.8248	0.6680	**0.0413**	0.8863	0.6945	0.6667
	Mean	1.0539	0.6668	**0.0134**	0.6239	0.6771	0.6152
	Std	0.8150	**2.5860 × 10^−4^**	0.0104	0.2949	0.0060	0.1572
*f* _6_	Best	2.6902 × 10^−6^	1.3140 × 10^−57^	1.9353 × 10^−9^	1.6776 × 10^−6^	**0**	**0**
	Worst	4.2475 × 10^−4^	5.2576 × 10^−50^	4.0498 × 10^−8^	2.1169 × 10^3^	**0**	**0**
	Mean	4.6277 × 10^−5^	1.9182 × 10^−51^	1.2558 × 10^−8^	470.0765	**0**	**0**
	Std	7.5722 × 10^−5^	9.5710 × 10^−51^	8.6975 × 10^−9^	554.1585	**0**	**0**
*f* _7_	Best	2.8302	4.8744 × 10^−4^	35.6262	23.8437	**0**	**0**
	Worst	11.6978	11.8819	69.5916	37.7208	**0**	**0**
	Mean	7.1201	1.7212	53.5446	32.1146	**0**	**0**
	Std	1.9084	2.5014	8.4248	2.8659	**0**	**0**
*f* _8_	Best	2.3485 × 10^−5^	1.2698 × 10^−47^	8.4305 × 10^−7^	2.6316 × 10^−5^	**0**	**0**
	Worst	5.9154 × 10^−4^	3.8306 × 10^−39^	3.6912 × 10^−6^	0.0019	**0**	**0**
	Mean	1.3335 × 10^−4^	1.4723 × 10^−40^	2.1448 × 10^−6^	4.9365 × 10^−4^	**0**	**0**
	Std	1.1816 × 10^−4^	6.9630 × 10^−40^	6.7603 × 10^−7^	4.5878 × 10^−4^	**0**	**0**
*f* _9_	Best	0.0214	5.0059 × 10^−4^	0.1207	0.0633	3.8672 × 10^−6^	**6.7322 × 10^−8^**
	Worst	0.0662	0.0063	0.2208	0.9340	1.2345 × 10^−4^	**1.3725 × 10^−5^**
	Mean	0.0388	0.0022	0.1618	0.4880	4.4905 × 10^−5^	**5.4986 × 10^−6^**
	Std	0.0107	0.0014	0.0249	0.2360	2.8655 × 10^−5^	**3.7638 × 10^−6^**
*f* _10_	Best	25,242	**0**	**0**	14,420	**0**	**0**
	Worst	56,647	**0**	**0**	19,647	**0**	**0**
	Mean	4.1697 × 10^4^	**0**	**0**	1.7400 × 10^4^	**0**	**0**
	Std	8.7103 × 10^3^	**0**	**0**	1.4435 × 10^3^	**0**	**0**
*f* _11_	Best	6.5543 × 10^−110^	**0**	2.4514 × 10^−17^	0.2369	**0**	**0**
	Worst	1.5207 × 10^−92^	**0**	5.3306 × 10^−10^	3.4087 × 10^5^	**0**	**0**
	Mean	7.7255 × 10^−94^	**0**	5.5002 × 10^−11^	2.9144 × 10^4^	**0**	**0**
	Std	2.9853 × 10^−93^	**0**	1.2907 × 10^−10^	7.8180 × 10^4^	**0**	**0**
*f* _12_	Best	7.4366	4.0903 × 10^−10^	178.9370	2.9405 × 10^−10^	**0**	**0**
	Worst	25.2881	0.0026	297.2858	1.3546 × 10^−6^	**0**	**0**
	Mean	14.4849	1.6588 × 10^−4^	258.7280	1.5486 × 10^−7^	**0**	**0**
	Std	5.1486	5.2388 × 10^−4^	26.6593	2.7336 × 10^−7^	**0**	**0**
*f* _13_	Best	6.8986 × 10^−7^	**0**	9.5128 × 10^−11^	341.9474	**0**	**0**
	Worst	0.0418	0.0317	2.7115 × 10^−6^	548.6863	**0**	**0**
	Mean	0.0073	0.0011	1.1044 × 10^−7^	453.6446	**0**	**0**
	Std	0.0110	0.0058	4.9356 × 10^−7^	50.0682	**0**	**0**
*f* _14_	Best	24.9001	**0**	2.6751 × 10^−10^	3.8881 × 10^−11^	**0**	**0**
	Worst	72.6456	**0**	0.9950	1.6855 × 10^−4^	**0**	**0**
	Mean	42.2098	**0**	0.0334	1.2826 × 10^−5^	**0**	**0**
	Std	9.7365	**0**	0.1816	3.4642 × 10^−5^	**0**	**0**
*f* _15_	Best	2.7359 × 10^−4^	4.4409 × 10^−15^	4.6745 × 10^−6^	2.3668 × 10^−7^	**8.8818 × 10^−16^**	**8.8818 × 10^−16^**
	Worst	0.0087	7.9936 × 10^−15^	2.0602 × 10^−5^	4.5716 × 10^−5^	**8.8818 × 10^−16^**	**8.8818 × 10^−16^**
	Mean	0.0012	5.1514 × 10^−15^	1.1813 × 10^−5^	9.9561 × 10^−6^	**8.8818 × 10^−16^**	**8.8818 × 10^−16^**
	Std	0.0019	1.4454 × 10^−15^	4.5981 × 10^−6^	1.0266 × 10^−5^	**0**	**0**
*f* _16_	Best	0.0036	2.2516 × 10^−10^	0.0197	1.9804 × 10^−9^	**0**	**0**
	Worst	0.0501	0.0344	0.0662	0.0513	**0**	**0**
	Mean	0.0158	0.0029	0.0398	0.0024	**0**	**0**
	Std	0.0094	0.0064	0.0116	0.0099	**0**	**0**
*f* _17_	Best	4.4072 × 10^−5^	3.6124 × 10^−43^	3.1203 × 10^−5^	9.5533 × 10^−7^	**0**	**0**
	Worst	0.0026	0.0135	0.0016	1.1938 × 10^−4^	**0**	**0**
	Mean	5.4095 × 10^−4^	4.4862 × 10^−4^	2.7629 × 10^−4^	1.8260 × 10^−5^	**0**	**0**
	Std	6.0504 × 10^−4^	0.0025	3.6245 × 10^−4^	2.4140 × 10^−5^	**0**	**0**
The number of optimal values	—	0	3	3	0	**14**	**15**

**Table 6 biomimetics-08-00210-t006:** The experimental results of several algorithms on the 17 test functions with Dim = 100.

Functions	Results	PSO	CSO	ABC	AFSA	ASCSO-S [28]	ADPCCSO
*f* _1_	Best	331.6384	3.8575 × 10^−10^	4.6681 × 10^−4^	1.0976 × 10^5^	**0**	**0**
	Worst	1.5637 × 10^3^	0.3189	0.0115	1.3888 × 10^5^	**0**	**0**
	Mean	820.8795	0.0231	0.0037	1.2477 × 10^5^	**0**	**0**
	Std	306.5691	0.0724	0.0030	7.3483 × 10^3^	**0**	**0**
*f* _2_	Best	1.3001 × 10^−9^	5.4401 × 10^−118^	8.9306 × 10^−11^	8.8725 × 10^−15^	**0**	**0**
	Worst	2.9796 × 10^−7^	1.1019 × 10^−15^	1.7231 × 10^−6^	1.3526 × 10^−7^	**0**	**0**
	Mean	3.9194 × 10^−8^	3.6821 × 10^−17^	1.5964 × 10^−7^	6.9148 × 10^−9^	**0**	**0**
	Std	5.6393 × 10^−8^	2.0117 × 10^−16^	3.2834 × 10^−7^	2.5352 × 10^−8^	**0**	**0**
*f* _3_	Best	193.9070	2.8194 × 10^−8^	5.0620 × 10^−4^	1.3041 × 10^−12^	**0**	**0**
	Worst	607.9061	16.5860	0.0080	4.3909 × 10^−7^	**0**	**0**
	Mean	359.4745	0.5679	0.0020	6.8692 × 10^−8^	**0**	**0**
	Std	119.4655	3.0256	0.0019	1.1022 × 10^−7^	**0**	**0**
*f* _4_	Best	1.7150 × 10^3^	1.7196 × 10^3^	**10.3203**	97.9925	98.4232	96.9640
	Worst	5.2518 × 10^3^	1.6970 × 10^5^	163.3205	97.9974	98.6505	**97.7413**
	Mean	3.1415 × 10^3^	9.3677 × 10^4^	**64.7252**	97.9948	98.5347	97.3542
	Std	969.8680	4.9125 × 10^4^	48.8932	**0.0013**	0.0542	0.1748
*f* _5_	Best	8.8700 × 10^3^	5.8244 × 10^3^	2.5520	0.2588	0.6697	**0.1650**
	Worst	4.6263 × 10^4^	5.8722 × 10^4^	32.2056	0.9957	0.7130	**0.6670**
	Mean	1.7986 × 10^4^	2.6486 × 10^4^	19.4081	0.6788	0.6824	**0.2483**
	Std	9.9693 × 10^3^	1.5023 × 10^4^	6.8118	0.3652	**0.0100**	0.1220
*f* _6_	Best	7.1656 × 10^3^	1.0137 × 10^−8^	0.0158	1.4010 × 10^6^	**0**	**0**
	Worst	2.7522 × 10^4^	0.4670	0.5992	1.7946 × 10^6^	**0**	**0**
	Mean	1.4753 × 10^4^	0.0374	0.1189	1.5276 × 10^6^	**0**	**0**
	Std	5.2502 × 10^3^	0.1085	0.1147	7.8348 × 10^4^	**0**	**0**
*f* _7_	Best	60.4254	20.0019	89.5451	64.4002	**0**	**0**
	Worst	74.0098	33.0724	94.9584	67.8016	**0**	**0**
	Mean	69.0845	27.6541	92.6977	66.4129	**0**	**0**
	Std	2.7611	3.0137	1.5126	0.8293	**0**	**0**
*f* _8_	Best	7.9422	2.4244 × 10^−25^	0.0434	1.8359 × 10^−5^	**0**	**0**
	Worst	31.8309	1.3145 × 10^−15^	1.8336	7.7035 × 10^−4^	**0**	**0**
	Mean	15.9717	1.8973 × 10^−16^	0.4174	2.4555 × 10^−4^	**0**	**0**
	Std	5.2235	2.8673 × 10^−16^	0.4529	2.1085 × 10^−4^	**0**	**0**
*f* _9_	Best	0.0132	0.8490	0.8410	0.0556	**2.9430 × 10^−6^**	5.9305 × 10^−6^
	Worst	0.0596	6.9853	1.9323	1.0584	1.8325 × 10^−4^	**1.6123 × 10^−4^**
	Mean	0.0389	2.7873	1.5058	0.3994	5.9754 × 10^−5^	**5.0547 × 10^−5^**
	Std	0.0121	1.4489	0.2431	0.2476	**4.0570 × 10^−5^**	4.0937 × 10^−5^
*f* _10_	Best	178,077	**0**	**0**	131,449	**0**	**0**
	Worst	245,665	3259	11	150,603	**0**	**0**
	Mean	2.0967 × 10^5^	208.3000	3.9000	1.4119 × 10^5^	**0**	**0**
	Std	1.6501 × 10^4^	608.1096	2.7959	5.5073 × 10^3^	**0**	**0**
*f* _11_	Best	1.5426 × 10^−104^	**0**	0.0022	3.2837 × 10^−4^	**0**	**0**
	Worst	2.4719 × 10^−91^	**0**	203.0441	2.0208 × 10^5^	**0**	**0**
	Mean	8.3044 × 10^−93^	**0**	20.6829	1.0851 × 10^4^	**0**	**0**
	Std	4.5118 × 10^−92^	**0**	43.5221	3.8186 × 10^4^	**0**	**0**
*f* _12_	Best	667.4600	57.7245	1.2511 × 10^3^	1.3428 × 10^−9^	**0**	**0**
	Worst	945.0711	201.4214	1.5822 × 10^3^	4.1973 × 10^−6^	**0**	**0**
	Mean	795.7159	121.6425	1.4465 × 10^3^	3.3379 × 10^−7^	**0**	**0**
	Std	71.2566	31.6731	74.6356	8.8407 × 10^−7^	**0**	**0**
*f* _13_	Best	5.3691	4.6892 × 10^−8^	6.4557 × 10^−4^	1.9224 × 10^3^	**0**	**0**
	Worst	15.8932	0.8468	0.1507	2.2859 × 10^3^	**0**	**0**
	Mean	8.3992	0.0971	0.0393	2.1567 × 10^3^	**0**	**0**
	Std	2.3010	0.2474	0.0401	72.4208	**0**	**0**
*f* _14_	Best	360.5859	5.4570 × 10^−12^	44.0741	6.6412 × 10^−9^	**0**	**0**
	Worst	554.4953	7.1574 × 10^−6^	94.4551	6.2310 × 10^−6^	**0**	**0**
	Mean	457.6672	3.5819 × 10^−7^	76.2198	9.6717 × 10^−7^	**0**	**0**
	Std	53.2555	1.3432 × 10^−6^	11.8763	1.5725 × 10^−6^	**0**	**0**
*f* _15_	Best	5.1581	2.6698 × 10^−4^	2.7054	8.8592	**8.8818 × 10^−16^**	**8.8818 × 10^−16^**
	Worst	7.3260	8.0990	4.3356	11.3301	**8.8818 × 10^−16^**	**8.8818 × 10^−16^**
	Mean	6.1988	4.2376	3.3019	10.4007	**8.8818 × 10^−16^**	**8.8818 × 10^−16^**
	Std	0.6392	3.0389	0.3805	0.6789	**0**	**0**
*f* _16_	Best	222.6462	11.7000	0.4054	5.5729 × 10^−10^	**0**	**0**
	Worst	430.1541	356.0054	5.2618	3.7702 × 10^−6^	**0**	**0**
	Mean	300.4803	124.7637	1.0924	3.6632 × 10^−7^	**0**	**0**
	Std	57.0504	80.3793	0.8972	7.3156 × 10^−7^	**0**	**0**
*f* _17_	Best	4.1014	5.5199 × 10^−16^	0.3310	1.5395 × 10^−6^	**0**	**0**
	Worst	15.7794	0.3483	2.8968	1.4224 × 10^−4^	**0**	**0**
	Mean	9.9203	0.1021	1.7783	3.1874 × 10^−5^	**0**	**0**
	Std	3.0857	0.0795	0.6698	3.4611 × 10^−5^	**0**	**0**
The number of optimal values	—	0	1	1	0	14	**16**

**Table 7 biomimetics-08-00210-t007:** The experimental results of several algorithms on the 17 test functions with Dim = 500.

Functions	Results	PSO	CSO	ABC	AFSA	ASCSO-S [28]	ADPCCSO
*f* _1_	Best	3.4417 × 10^5^	2.5543 × 10^3^	3.6768 × 10^5^	1.0969 × 10^6^	**0**	**0**
	Worst	4.6704 × 10^5^	6.8327 × 10^4^	4.4583 × 10^5^	1.1766 × 10^6^	**0**	**0**
	Mean	3.8680 × 10^5^	2.5851 × 10^4^	4.1182 × 10^5^	1.1455 × 10^6^	**0**	**0**
	Std	2.8825 × 10^4^	2.1362 × 10^4^	2.1084 × 10^4^	1.7879 × 10^4^	**0**	**0**
*f* _2_	Best	2.9636 × 10^−5^	2.0082 × 10^−20^	0.0013	3.8515 × 10^−24^	**0**	**0**
	Worst	5.8766 × 10^−4^	1.0013 × 10^−8^	0.0954	6.4414 × 10^−21^	**0**	**0**
	Mean	2.4724 × 10^−4^	3.4055 × 10^−10^	0.0283	1.1455 × 10^−21^	**0**	**0**
	Std	1.5991 × 10^−4^	1.8271 × 10^−9^	0.0222	1.7334 × 10^−21^	**0**	**0**
*f* _3_	Best	6.8732 × 10^5^	1.5062 × 10^3^	8.3936 × 10^5^	7.8388 × 10^−11^	**0**	**0**
	Worst	9.1084 × 10^5^	1.5161 × 10^5^	1.1140 × 10^6^	8.8722 × 10^−7^	**0**	**0**
	Mean	7.9392 × 10^5^	4.7686 × 10^4^	9.8574 × 10^5^	1.3798 × 10^−7^	**0**	**0**
	Std	5.4605 × 10^4^	3.9801 × 10^4^	6.4256 × 10^4^	2.4515 × 10^−7^	**0**	**0**
*f* _4_	Best	1.7194 × 10^6^	1.3918 × 10^6^	6.3634 × 10^6^	493.9575	496.7487	**493.9299**
	Worst	3.5606 × 10^6^	1.9726 × 10^6^	1.2479 × 10^7^	493.9587	497.2473	**493.9573**
	Mean	2.3993 × 10^6^	1.6391 × 10^6^	9.3641 × 10^6^	493.9579	496.9662	**493.9538**
	Std	3.9854 × 10^5^	1.3493 × 10^5^	1.4413 × 10^6^	**2.6809 × 10^−4^**	0.1279	0.0061
*f* _5_	Best	1.0013 × 10^8^	1.8404 × 10^6^	7.8591 × 10^7^	0.3184	0.6689	**0.2500**
	Worst	1.3784 × 10^8^	7.8611 × 10^6^	1.5676 × 10^8^	0.9999	**0.7017**	1.000
	Mean	1.1896 × 10^8^	2.8855 × 10^6^	1.2822 × 10^8^	0.9772	0.6793	**0.4515**
	Std	1.0854 × 10^7^	1.0721 × 10^6^	1.7802 × 10^7^	0.1244	**0.0080**	0.3363
*f* _6_	Best	2.9002 × 10^7^	2.2390 × 10^5^	3.5757 × 10^7^	9.9591 × 10^7^	**0**	**0**
	Worst	3.5667 × 10^7^	6.8730 × 10^6^	4.6993 × 10^7^	1.0592 × 10^8^	**0**	**0**
	Mean	3.3281 × 10^7^	2.7459 × 10^6^	4.1792 × 10^7^	1.0295 × 10^8^	**0**	**0**
	Std	1.7553 × 10^6^	1.9846 × 10^6^	2.6048 × 10^6^	1.5075 × 10^6^	**0**	**0**
*f* _7_	Best	86.9503	28.1422	98.3712	85.7017	**0**	**0**
	Worst	92.0266	37.3954	99.4773	86.6041	**0**	**0**
	Mean	90.0683	32.4090	99.0033	86.0787	**0**	**0**
	Std	1.0962	2.6020	0.3065	0.1978	**0**	**0**
*f* _8_	Best	—	7.2696 × 10^−5^	14.8251	2.5230 × 10^−6^	**0**	**0**
	Worst	—	0.0272	35.1057	6.2084 × 10^−4^	**0**	**0**
	Mean	—	0.0038	23.2632	1.6534 × 10^−4^	**0**	**0**
	Std	—	0.0052	5.8171	1.4938 × 10^−4^	**0**	**0**
*f* _9_	Best	6.8570 × 10^3^	109.6845	6.4870 × 10^3^	0.0405	**4.3034 × 10^−6^**	4.3945 × 10^−6^
	Worst	9.8138 × 10^3^	248.1032	1.1307 × 10^4^	1.0915	2.5897 × 10^−4^	**2.3338 × 10^−4^**
	Mean	8.0498 × 10^3^	165.4784	8.7373 × 10^3^	0.4747	8.6590 × 10^−5^	**7.8670 × 10^−5^**
	Std	713.7037	35.6537	1.3277 × 10^3^	0.3269	6.7364 × 10^−5^	**6.1413 × 10^−5^**
*f* _10_	Best	1,184,738	696	338,407	1,090,299	**0**	**0**
	Worst	1,420,609	78,661	438,699	1,178,878	**0**	**0**
	Mean	1.3025 × 10^6^	3.2412 × 10^4^	4.0323 × 10^5^	1.1436 × 10^6^	**0**	**0**
	Std	5.1289 × 10^4^	2.1558 × 10^4^	2.1198 × 10^4^	2.1114 × 10^4^	**0**	**0**
*f* _11_	Best	2.2971 × 10^−108^	**0**	238.1984	2.5935	**0**	**0**
	Worst	2.3336 × 10^−90^	**0**	3.2393 × 10^8^	6.8893 × 10^12^	**0**	**0**
	Mean	8.4855 × 10^−92^	**0**	2.4048 × 10^7^	3.5147 × 10^11^	**0**	**0**
	Std	4.2637 × 10^−91^	**0**	6.8332 × 10^7^	1.3599 × 10^12^	**0**	**0**
*f* _12_	Best	7.4466 × 10^3^	423.5607	8.0224 × 10^3^	7.4597 × 10^−10^	**0**	**0**
	Worst	9.0534 × 10^3^	957.8956	9.0259 × 10^3^	7.0508 × 10^−5^	**0**	**0**
	Mean	8.3756 × 10^3^	605.4779	8.5640 × 10^3^	4.8219 × 10^−6^	**0**	**0**
	Std	411.8127	134.3923	225.0238	1.5936 × 10^−5^	**0**	**0**
*f* _13_	Best	3.1602 × 10^3^	4.1279	3.3720 × 10^3^	1.2415 × 10^4^	**0**	**0**
	Worst	4.0106 × 10^3^	619.8709	4.1088 × 10^3^	1.3397 × 10^4^	**0**	**0**
	Mean	3.5174 × 10^3^	257.6846	3.7513 × 10^3^	1.2997 × 10^4^	**0**	**0**
	Std	207.0651	196.7478	178.7140	245.8616	**0**	**0**
*f* _14_	Best	4.4677 × 10^3^	0.0554	3.4416 × 10^3^	0	**0**	**0**
	Worst	5.3152 × 10^3^	28.4260	3.9138 × 10^3^	9.2223 × 10^−6^	**0**	**0**
	Mean	4.9007 × 10^3^	1.9175	3.7437 × 10^3^	5.3898 × 10^−7^	**0**	**0**
	Std	223.7973	5.1479	122.1901	1.6800 × 10^−6^	**0**	**0**
*f* _15_	Best	19.9067	9.3629	18.7378	19.4867	**8.8818 × 10^−16^**	**8.8818 × 10^−16^**
	Worst	20.3297	10.8053	19.1688	19.6899	**8.8818 × 10^−16^**	**8.8818 × 10^−16^**
	Mean	20.0874	10.1880	18.9621	19.6119	**8.8818 × 10^−16^**	**8.8818 × 10^−16^**
	Std	0.0956	0.2990	0.1058	0.0535	**0**	**0**
*f* _16_	Best	3.6786 × 10^4^	5.9190 × 10^3^	5.8397 × 10^3^	2.4219 × 10^−13^	**0**	**0**
	Worst	5.9557 × 10^4^	1.1086 × 10^4^	6.9910 × 10^4^	2.3802 × 10^−7^	**0**	**0**
	Mean	4.8675 × 10^4^	7.6308 × 10^3^	4.6904 × 10^4^	2.7949 × 10^−8^	**0**	**0**
	Std	6.0856 × 10^3^	1.0778 × 10^3^	1.4820 × 10^4^	5.2233 × 10^−8^	**0**	**0**
*f* _17_	Best	579.0176	0.3822	347.2300	6.4884 × 10^−7^	**0**	**0**
	Worst	846.7066	11.2094	455.8400	1.1283 × 10^−4^	**0**	**0**
	Mean	651.8188	2.8344	425.6886	3.0884 × 10^−5^	**0**	**0**
	Std	53.6387	2.4758	22.2323	2.7903 × 10^−5^	**0**	**0**
The number of optimal values	—	0	1	0	0	**14**	**17**

**Table 8 biomimetics-08-00210-t008:** Friedman test results of algorithms.

Algorithms	Average Ranking	Ranking
ADPCCSO	1.50	1
ASCSO-S	1.79	2
CSO	4.06	3
AFSA	4.24	4
ABC	4.24	4
PSO	5.18	5

**Table 9 biomimetics-08-00210-t009:** The experimental results of three improved CSO algorithms with Dim = 100.

Functions	Results	GCSO [27]	DMCSO [29]	ADPCCSO
*f* _1_	Best	1.85 × 10^−22^	5.2267 × 10^−14^	**0**
	Worst	8.95 × 10^−22^	1.0984 × 10^−2^	**0**
	Mean	3.44 × 10^−22^	5.8629 × 10^−4^	**0**
	Std	1.49 × 10^−22^	2.0933 × 10^−3^	**0**
*f* _2_	Best	—	2.0470 × 10^−52^	**0**
	Worst	—	3.6166 × 10^−16^	**0**
	Mean	—	1.5245 × 10^−17^	**0**
	Std	—	6.7682 × 10^−17^	**0**
*f* _3_	Best	—	—	**0**
	Worst	—	—	**0**
	Mean	—	—	**0**
	Std	—	—	**0**
*f* _4_	Best	98.4	**2.1671 × 10 ^−5^**	96.9640
	Worst	99.1	**15.966**	97.7413
	Mean	98.4	**1.1912**	97.3542
	Std	**0.1685**	3.2648	0.1748
*f* _5_	Best	—	0.23433	**0.1650**
	Worst	—	0.95006	**0.6670**
	Mean	—	0.39495	**0.2483**
	Std	—	0.19341	**0.1220**
*f* _6_	Best	—	—	**0**
	Worst	—	—	**0**
	Mean	—	—	**0**
	Std	—	—	**0**
*f* _7_	Best	—	—	**0**
	Worst	—	—	**0**
	Mean	—	—	**0**
	Std	—	—	**0**
*f* _8_	Best	—	1.4932 × 10^−29^	**0**
	Worst	—	1.3772 × 10^−25^	**0**
	Mean	—	1.1269 × 10^−26^	**0**
	Std	—	2.7019 × 10^−26^	**0**
*f* _9_	Best	—	—	5.9305 × 10^−6^
	Worst	—	—	1.6123 × 10^−4^
	Mean	—	—	5.0547 × 10^−5^
	Std	—	—	4.0937 × 10^−5^
*f* _10_	Best	—	—	**0**
	Worst	—	—	**0**
	Mean	—	—	**0**
	Std	—	—	**0**
*f* _11_	Best	2.71 × 10^−90^	5.0821 × 10^−16^	**0**
	Worst	4.94 × 10^−75^	7.8047 × 10^−4^	**0**
	Mean	1.23 × 10^−75^	1.1682 × 10^−4^	**0**
	Std	1.89 × 10^−75^	2.0372 × 10^−4^	**0**
*f* _12_	Best	—	2.4456 × 10^−6^	**0**
	Worst	—	3.1745 × 10^2^	**0**
	Mean	—	1.2118 × 10^2^	**0**
	Std	—	1.1470 × 10^2^	**0**
*f* _13_	Best	**0**	1.0436 × 10^−13^	**0**
	Worst	3.33 × 10^−16^	1.5270 × 10^−4^	**0**
	Mean	2.78 × 10^−17^	1.5162 × 10^−5^	**0**
	Std	6.16 × 10^−17^	3.8028 × 10^−5^	**0**
*f* _14_	Best	**0**	2.2612 × 10^−13^	**0**
	Worst	1.95 × 10^−14^	8.8588 × 10^−6^	**0**
	Mean	2.72 × 10^−15^	6.6629 × 10^−7^	**0**
	Std	5.04 × 10^−15^	2.0084 × 10^−6^	**0**
*f* _15_	Best	5.21 × 10^−24^	1.3195 × 10^−7^	**8.8818 × 10^−16^**
	Worst	1.69 × 10^−21^	9.2963 × 10^−2^	**8.8818 × 10^−16^**
	Mean	9.08 × 10^−24^	9.5407 × 10^−3^	**8.8818 × 10^−16^**
	Std	2.44 × 10^−24^	2.2479 × 10^−2^	**0**
*f* _16_	Best	—	4.5274 × 10^−5^	**0**
	Worst	—	7.6947 × 10^−2^	**0**
	Mean	—	1.1517 × 10^−2^	**0**
	Std	—	1.9620 × 10^−2^	**0**
*f* _17_	Best	—	9.5471 × 10^−30^	**0**
	Worst	—	9.2151 × 10^−2^	**0**
	Mean	—	9.2379 × 10^−3^	**0**
	Std	—	1.9493 × 10^−2^	**0**

**Table 10 biomimetics-08-00210-t010:** The experimental results of the ADPCCSO algorithms with Dim = 1000.

Functions	Mean	std	Best	Worst
*f* _1_	0	0	0	0
*f* _2_	0	0	0	0
*f* _3_	0	0	0	0
*f* _4_	988.9073	4.2390 × 10^−4^	988.9067	988.9089
*f* _5_	0.8900	0.2525	0.2505	1.0000
*f* _6_	0	0	0	0
*f* _7_	0	0	0	0
*f* _8_	0	0	0	0
*f* _9_	7.4907 × 10^−5^	7.1055 × 10^−5^	3.3858 × 10^−7^	2.8846 × 10^−4^
*f* _10_	0	0	0	0
*f* _11_	0	0	0	0
*f* _12_	0	0	0	0
*f* _13_	0	0	0	0
*f* _14_	0	0	0	0
*f* _15_	8.8818 × 10^−16^	0	8.8818 × 10^−16^	8.8818 × 10^−16^
f_16_	0	0	0	0
*f* _17_	0	0	0	0

**Table 11 biomimetics-08-00210-t011:** The observed growth concentration of glutamate.

Time (h)	Concentration (g/L)	Time (h)	Concentration (g/L)
2	0.321	12	0.869
3	0.353	13	0.878
4	0.369	14	0.879
5	0.408	15	0.893
6	0.581	16	0.894
7	0.640	17	0.900
8	0.742	18	0.901
9	0.781	19	0.902
10	0.824	20	0.903
11	0.855	21	0.903

**Table 12 biomimetics-08-00210-t012:** The experimental results of optimal solutions obtained by various algorithms.

	Parameters	α	β	γ	δ
Algorithms					
VS-FOA [36]		0.8965	4.8369	0.6079	3.0260
ASCSO-S [28]		0.8973	5.5	0.6556	3.6327
ADPCCSO		0.8949	6.5522	0.7533	4.4263

**Table 13 biomimetics-08-00210-t013:** The growth concentration of glutamate predicted by each algorithm.

Time (h)	VS-FOA [36]	ASCSO-S [28]	ADPCCSO
2	0.2686	0.2821	0.2858
3	0.3260	0.3366	0.3383
4	0.3935	0.4003	0.3997
5	0.4705	0.4731	0.4705
6	0.5542	0.5534	0.5499
7	0.6388	0.6363	0.6341
8	0.7161	0.7142	0.7155
9	0.7789	0.7792	0.7840
10	0.8244	0.8265	0.8328
11	0.8543	0.8572	0.8626
12	0.8725	0.8754	0.8789
13	0.8831	0.8856	0.8872
14	0.8891	0.8912	0.8912
15	0.8924	0.8941	0.8932
16	0.8943	0.8956	0.8941
17	0.8953	0.8964	0.8945
18	0.8958	0.8968	0.8947
19	0.8961	0.8971	0.8948
20	0.8963	0.8972	0.8949
21	0.8964	0.8972	0.8949
fit	0.0097	0.0089	0.0087

**Table 14 biomimetics-08-00210-t014:** The comparison results of three algorithms.

	Algorithms	VS-FOA	ASCSO-S	ADPCCSO
Indexes				
RMSE		0.0220	0.0211	**0.0209**
MAE		0.0136	**0.0135**	0.0146
R^2^		0.9888	0.9896	**0.9899**

## Data Availability

All data used to support the findings of this study are included within the article. Color versions of all figures in this paper are available from the corresponding author upon request.

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
