# Peer review of "An Adaptive Dual-Population Collaborative Chicken Swarm Optimization Algorithm for High-Dimensional Optimization"

_biomimetics, 2023, doi:10.3390/biomimetics8020210_

Round 1

Reviewer 1 Report (Previous Reviewer 3)

The work fits the journal's focus, and readers will be interested. The study has specific strengths, but I think it is faulty from various angles, so it must be updated.

In my opinion, the introduction has to be strengthened to more clearly state the work's purpose and the technique's features. It's unclear how it operates as it stands. The author should also make it apparent how this work advances the state of the art, emphasizing its originality......

Concerning the literature review, each paper should specify the proposed methodology, novelty, and experimental results clearly. At the end of related works, emphasize more clearly in a few lines what overall technical gaps were discovered in previous works that led to the design of the proposed approach. You can use the following papers as references better to define the context and the various possible solutions: https://www.sciencedirect.com/science/article/pii/S0167739X08000472 and https://ieeexplore.ieee.org/abstract/document/9933766.

The experimental part needs some clarity. More details are necessary. It is necessary to enhance the analysis and interpretation of the test results. In order to support the suggested thesis, the findings section should be connected to a discussion part.

The methodology's future application should be expanded upon or highlighted. Clarify the article's conclusion and its importance for further study as you improve it.

Author Response

The work fits the journal's focus, and readers will be interested. The study has specific strengths, but I think it is faulty from various angles, so it must be updated.

In my opinion, the introduction has to be strengthened to more clearly state the work's purpose and the technique's features. It's unclear how it operates as it stands. The author should also make it apparent how this work advances the state of the art, emphasizing its originality......

Our response: Thanks for your comments. Maybe our expression is not clear enough. We have specifically improved our paper. At the same time, in order to present them more clearly to the reviewers, we have deliberately listed them as follows.

The work's purpose:

With the increase of the dimensions of the optimization problems, the CSO algorithm is prone to premature convergence. Therefore, for the optimization problem of high-dimensional complex functions, Yang et al. constructed a genetic CSO algorithm by introducing the idea of genetic algorithm into the CSO algorithm, and verified the performance of the proposed algorithm on 10 benchmark functions[25]. Although the convergence speed and stability have been improved, the solution accuracy is still unsatisfactory. Gu et al. realized the solution of high-dimensional complex function optimization problems by removing the chicks in the chicken swarm and introducing an inverted S-shaped inertial weight to construct an adaptive simplified CSO algorithm[26]. Although the proposed algorithm is significantly better than some other algorithms in solution accuracy, there is still room for improvement in convergence speed. By introducing the dissipative structure and differential mutation operation into the basic CSO algorithm, Han constructed a hybrid CSO algorithm to avoid the premature convergence in solving high-dimensional complex problems, and verified the performance of the proposed algorithm on 18 standard functions[27]. Although its convergence performance has been improved, the solution accuracy should be further enhanced.

To address the aforementioned issues, we propose an adaptive dual-population collaborative CSO (ADPCCSO) algorithm in this paper. The algorithm solves the high-dimensional complex problems by using an adaptive adjustment strategy for parameter G, an improvement strategy for foraging behaviors and a dual-population collaborative optimization strategy.

The technique's features and its originality :

Specifically, the main technical features and originality of this paper are given below.

(1) The value of parameter G is given by an adaptive dynamic adjustment method, so as to balance the breadth and depth search abilities of the algorithm.

(2) To improve the solution accuracy and depth optimization ability of CSO algorithm, an improvement strategy for foraging behaviors is proposed by introducing an improvement factor and adding a kind of chick's foraging behavior near the optimal value.

(3) A dual-population collaborative optimization strategy based on the chicken swarm and artificial fish swarm is constructed to enhance the global search ability of the whole algorithm.

The simulation experiments on the selected standard test functions and the parameter estimation problem of Richards model show that the ADPCCSO algorithm is better than some other meta-heuristic optimization algorithms in terms of solution accuracy and convergence performance etc.

Concerning the literature review, each paper should specify the proposed methodology, novelty, and experimental results clearly. At the end of related works, emphasize more clearly in a few lines what overall technical gaps were discovered in previous works that led to the design of the proposed approach. You can use the following papers as references better to define the context and the various possible solutions: https://www.sciencedirect.com/science/article/pii/S0167739X08000472 and https://ieeexplore.ieee.org/abstract/document/9933766.
Our response: Thanks for your comments. For this problem, we have added some key references in this paper such as [11] and [12] etc. and discuss them in Section 1, which are listed in part as follows.

Meta-heuristic optimization algorithm is a class of random search algorithm proposed by simulating biological intelligence in nature[8], which has been successfully applied in various fields, such as the internet of Things[9], network information systems[10, 11], multi-robot space exploration[12], and so on.

The experimental part needs some clarity. More details are necessary. It is necessary to enhance the analysis and interpretation of the test results. In order to support the suggested thesis, the findings section should be connected to a discussion part.

Our response: Thanks for your comments. Maybe our expression is not clear enough. We have specifically improved our paper. At the same time, in order to present them more clearly to the reviewers, we have deliberately listed them as follows. Some key explanations are marked in red.

The analysis and interpretation of the test results of Section 4.2:

“It can be seen from Table 3 that the optimization results of ACSO and DCCSO algorithms on almost all benchmark test functions are far superior to those of CSO algorithm (Only on function f2, the optimization results of DCCSO algorithm are slightly inferior to those of CSO algorithm), especially the experimental data on functions f10 and f11 have reached the theoretical optimal values. It shows the effectiveness of the two improvement strategies proposed in Sections 3.1 and 3.3 in the aspect of solution accuracy.”

“As can be seen from Fig. 2, the convergence performance of both ACSO and DCCSO algorithms is significantly superior to that of CSO algorithm, which proves the effectiveness of the two improvement strategies proposed in this paper in the aspect of convergence performance.”

The analysis and interpretation of the test results of Section 4.3:

“As can be seen from Table 4, the ADPCCSO-[32] only obtained optimal values on 5 functions, while the ADPCCSO algorithm obtained optimal values on 16 functions, where the theoretical optimal values were obtained on 13 functions. Only on function f5, the results of ADPCCSO algorithm are slightly inferior to those of the ADPCCSO-[32]. It shows the effectiveness of the improvement strategy proposed in Section 3.2 in terms of solution accuracy.”

“It is obvious from Fig. 3 that the convergence performance of ADPCCSO algorithm is better than that of ADPCCSO-[32] as a whole. Especially on function f15, ADPCCSO algorithm has more obvious advantages in convergence performance, and it began to converge stably around the 18th generation.”

The analysis and interpretation of the test results of Section 4.4:

It is not difficult to see from Tables 5-7 that ADPCCSO and ASCSO-S algorithms are far superior to the other 4 swarm intelligence algorithms in terms of solution accuracy and stability. Among them, ADPCCSO algorithm has the best performance, especially when Dim=500, it has obtained the optimal values in all 17 functions, while the number of optimal results calculated by the ASCSO-S algorithm is 14. And on function f5, the operation results of ADPCCSO algorithm at Dim=100 and Dim=500 are far better than those at Dim=30, which also shows to a certain extent that the ADPCCSO algorithm is more suitable for handling higher-dimensional complex optimization problems.

As can be seen from Table 5, although ABC algorithm has obtained the optimal values in three functions, its optimization ability is getting worse and worse as the dimension of the problem increases. On the contrary, AFSA shows a higher optimization ability (When Dim=500, its optimization ability on 11 functions is much better than that of ABC algorithm), which is one of the reasons why we construct a dual-population collaborative optimization strategy based on the chicken swarm and artificial fish swarm to solve high-dimensional optimization problems. It is noteworthy that the operation results of PSO algorithm on function f8 are not given in Table 7. This is because when Dim=500, its fitness function values often exceed the maximum positive value that the computer can represent, resulting in the algorithm unable to obtain suitable operation results. This also shows that PSO algorithm is not suitable for handling the higher-dimensional complex optimization problems.

Summarize the reason why the solution accuracy of ADPCCSO and ASCSO-S algorithms is better than that of the other 5 algorithms. This may be due to the fact that both algorithms introduce an improvement factor (the latter is called inertial weight) into the position update formula of chicken swarm. The reason why the performance of the former in terms of solution accuracy is better than that of the latter may be because the former uses an improvement strategy for foraging behaviors, which not only improves the depth optimization ability of the algorithm, but also improves its solution accuracy.”

As can be seen from Fig. 4, ADPCCSO algorithm has the best convergence performance on 16 functions, only on function f4, its convergence is slightly inferior to that of ABC algorithm. ASCSO-S ranks second in the convergence performance, AFSA and CSO are tied for third place (This is another reason why we construct a dual-population collaborative optimization strategy based on chicken swarm and artificial fish swarm).

Summarize the reason why the convergence performance of ADPCCSO and ASCSO-S algorithms is better than that of the other 4 algorithms as a whole. This may be because both algorithms use adaptive dynamic adjustment strategies. The convergence performance of the former is superior to that of the latter, which may be due to the use of dual-population collaborative optimization strategy in the ADPCCSO algorithm, which improves the convergence performance of the algorithm. In addition, by carefully observing Fig. 4, it is not difficult to find that on functions f1-f3, f6-f8, f10-f14 and f16-f17, it seems that the convergence curves of ADPCCSO and ASCSO-S algorithms in the late iteration stage are not fully presented. This is because both algorithms have found the theoretical optimal value of 0 in these functions, and 0 has no logarithm.”

The analysis and interpretation of the test results of Section 4.6:

“In Table 9, GCSO[27] counted the operation results of 6 functions out of 17 test functions, but only obtained the optimal values on the standard deviation of function f4 and the best values of functions f13 and f14. DMCSO[29] counted the operation results of 12 functions out of 17 test functions, and only obtained the optimal values on function f4. However, the operation results of ADPCCSO algorithm are better than those of the above two algorithms overall. Only on function f4, the operation results of ADPCCSO algorithm are worse than those of DMCSO[29]. It shows the advantages of ADPCCSO algorithm.”

The analysis and interpretation of the test results of Section 4.7:

“As can be seen from Table 10, even when the dimension of the optimization problem is adjusted to 1000, the proposed algorithm can still achieve satisfactory optimization accuracy on most test functions, only on functions f4, f5 and f9, the experimental data fluctuate slightly. This indicates that when the dimension increases, the proposed algorithm will not be greatly affected, which fully demonstrates that the ADPCCSO algorithm still has a competitive advantage in dealing with higher-dimensional optimization problems.”

The methodology's future application should be expanded upon or highlighted. Clarify the article's conclusion and its importance for further study as you improve it.

Our response: Thanks for your comments. Maybe our expression is not clear enough. We have specifically improved our paper. At the same time, in order to present them more clearly to the reviewers, we have deliberately listed them as follows.

The methodology's future application:

Therefore, how to improve the performance of the algorithm to better solve more complex large-scale optimization problems still needs further research. Moreover, in the future research work, it is also a good choice to apply this algorithm to other fields, such as constrained optimization problem, multi-objective optimization problem and vehicle routing problem.

The article's conclusion and its importance for further study:

The experimental results preliminarily show that the proposed algorithm has obvious advantages over other comparison algorithms in terms of solution accuracy and convergence performance. This provides new ideas for the study of high-dimensional optimization problems.

Reviewer 2 Report (Previous Reviewer 1)

All my comments are properly adressed.

Author Response

Our response: Thanks for your comments.

Round 2

Reviewer 1 Report (Previous Reviewer 3)

The paper was improved therefore it can be accepted for the publication in its current form.

This manuscript is a resubmission of an earlier submission. The following is a list of the peer review reports and author responses from that submission.

Round 1

Reviewer 1 Report

In this paper, an adaptive dual-population collaborative chicken swarm optimization (ADPCCSO) algorithm is proposed to solve numerical optimization problems. Experiments are conducted on 17 benchmark functions. The ADPCCSO algorithm is compared to the bacterial foraging algorithm (BFA), artificial fish swarm algorithm (AFSA), artificial bee colony (ABC) and particle swarm optimization (PSO). The APDCCSO is also utilized in the parameter estimation problem of Richards model to further verify its performance.

This paper should be accepted subject to the following conditions:

There are a lot of novel hybrid metaheuristis which are not mentioned in the paper. Some examples are:

(1)    Jooda, J.O.; Makinde, B.O.; Odeniyi, O.A.; Okandeji, M.A. A review on hybrid artificial bee colony for feature selection. Glob. J.Adv. Res. 2021, 8, 170–177.

(2)    Brajević, I.; Stanimirović, P.S.; Li, S.; Cao, X.; Khan, A.T.; Kazakovtsev, L.A. Hybrid Sine Cosine Algorithm for Solving Engineering Optimization Problems. Mathematics 202210, 4555.

(3)    Robert Pellerin, Nathalie Perrier, François Berthaut, A survey of hybrid metaheuristics for the resource-constrained project scheduling problem, European Journal of Operational Research, Volume 280, Issue 2, 2020, Pages 395-416.

 -The computational complexity of the proposed approach has to be explained in the Section 3.4.

-A common recommendation for fair comparison amongst meta-heuristic algorithms is to compare algorithms based on an equal number of consumed fitness evaluations. Hence, in Section 4.1, the number of consumed fitness evaluations for ASCSO-S, ABC, AFSA, BFA, CSO, and PSO algorithms must be mentioned.  Also, the specific control parameter settings of these metaheuristic algorithms parameters must be explained in the paper.

-The advantages and the limitations of the proposed approach should be discussed with more details in the paper.

-The plans for future work should be elaborated with more details.

Reviewer 2 Report

The paper continuous with the recent trend of proposing new metaphor-based optimization algorithms and applying known concepts from other meta-heuristic studies to these new metaphor-inspired concepts. Most of these metaphor-inspired optimization algorithms do not present anything new other than the metaphor. When analyzing the underlying mathematical equations and algorithm, there are strong equivalences to that of established meta-heuristics. Therefore, these studies do not contribute anything new to the research domain.The authors are advised to read the papers by Christian Leonardo Camacho Villalón (see https://scholar.google.com/citations?user=kFuYHw4AAAAJ&hl=en).

The Chicken swarm optimization algorithm is no exception to this trend. The position update equations are very similar to that of PSO, with additional equivalences to differential evolution. It is therefore possible to reformulate these update equations in a general form, equivalent to a PSO.

Publications on new metaphor-based optimization algorithms and variations thereof only makes a contribution if the authors can clearly show that the underlying mathematical equations and search behaviors differ from established meta-heuristics.

Based on the above, my recommendation is that the paper be rejected. In addition, find below other concerns with the paper:

1. The paper refers to fish swarm optimization, and that this is used to develop an adaptive approach to the CSO. Yet, another metaphor-based optimization algorithm. However, detail of this fish swarm algorithm is not provided, and how it is used to produce an adaptive version of the CSO is not provided.

2. For the second paragraph of the introduction, give references to these algorithms. Also, there are over a thousand of these metaphor-based optimization algorithms. See http://fcampelo.github.io/EC-Bestiary/

3. Section 3.2: To refer to this as learning is not correct. There is no learning. In fact, the basic principles of swarm intelligence excludes learning behavior.

4. Equations 14 and 15: What are the justifications for the 2/3?

5. Lines 241-244: What do you mean by the breadth and depth of search ability? I guess that you mean the more standard terminology of exploration and exploitation?

6. Line 247: Why these control parameter settings?

7. Line 288: What is the rationale of running the algorithms for 100 and 1000 iterations?

8. With reference to table 2: Why these values for control parameters? Did you apply a tuning approaches to find best possible values for each of the algorithms? If not, then your comparisons cannot be considered fair.

9. A dimensionality of 100 is no longer considered high-dimensional. If you claim to solve high-dimensional problems, you need to consider various dimensionalities up to at least 1000.

10. Statistical tests have not been employed to test if differences in performance are statistically significant or not.

11. Table 4 and the following tables. I am sorry, but I simply do not believe the many 0 results entered.

Reviewer 3 Report

The paper is well-structured, interesting, and readable. However, the scientific contribution of the paper is unclear: the theoretical and practical implications are rather limited. Also, the technical challenges are quite unclear. Not enough technical details. It is not possible to replicate the works since some of the details of the proposed solution are not specified and cannot be guessed by the reader.

The abstract needs some attraction in terms of scientific contribution. It should be rewritten to highlight the research gap in existing research and in the literature. The importance of the proposed integrated approach with respect to the problem statement should have been in focus.

The section Introduction should clarify better and provide concise information with regard to the problem definition and scope of the paper. The contribution summarization should be remarked on better. Moreover, the connection between the problem and the solution proposed is also not clearly pointed out. Emphasize the novelty introduced.

The proposed methodology and what was found better as compared to existing ones (in the Introduction) should be highlighted better. 

Also, highlight the research gap in existing research and in literature. The importance of the proposed integrated approach with respect to the problem statement should have been in focus. Each paper should be explicit about the proposed methodology, novelty, and experimental findings in the literature review.

Highlight more clearly in a few lines what general technical shortcomings in previous works were found to have prompted the development of the suggested approach at the conclusion of related works. You can use the following papers as references to more clearly define the context and the various potential solutions: https://www.sciencedirect.com/science/article/abs/pii/S0957417421012598 and https://ieeexplore.ieee.org/abstract/document/4610103.

The methodology's future application should be expanded upon or highlighted. As you improve it, be sure to clarify the article's conclusion and its significance for future research. Enhance the discussion section where the study's findings and their practical applications are discussed (some of this is covered in the Conclusion section).

Round 2

Reviewer 1 Report

All my comments are properly addressed.

Reviewer 2 Report

My first concern with the paper remains, and I cannot provide a positive recommendation on yet another metaphor-based optimization algorithm if specific novelty of this algorithm is not clearly illustrated with reference to existing and established nature-inspired meta-heuristics. The similarities with differential evolution and particle swarm optimization make this not a different novel optimization algorithm. It is not a novel optimization algorithm, but a specialization of existing algorithms.

My comment 1: You have not addressed this comment. Ultimately you make use of the AFSA, and this is not described, and it remains the case that another non-novel metaphor-based optimization algorithm is used, where the components of this algorithm can be found in established algorithms.

Comment 2: My comment has been ignored. Also see Heuristics, 1:33–42, 1995.
[17] K. Hussain, MN. Mohd Salleh, S. Cheng, and Y. Shi. Metaheuristic
research: a comprehensive survey. Artificial Intelligence Review, 52(4):2191–2233, 2019.

Comment 3: Again, there is no learning. The authors need to study the definition of learning.

Comment 5: Where does breadth-first search and depth-first search fit into the optimization algorithm and process? I fail to see this.

Comment 6: Control parameters from literature can only be used if these control parameter values have been found as a result of a formal tuning process on the same optimization problems that you have used. If not, then the empirical comparisons cannot be considered fair. Also, detail of the trial and error process is needed.

Comment 7: You do not provide a reason for why you have used two different values for the number of iterations. Also, such comparison is only fair if all of the algorithms use the same swarm sizes.

Comment 8: See comment 6.

Comment 9: The authors have again decided to ignore one of my comments. Many recent papers on large-scale optimization consider large-dimensional problems.

Comment 10: For the Friedman test, did you apply a post-hoc test to account for the family-wise error rate, i.e. Type 1 errors?

Comment 11: What I am saying here is that this is not possible for all of these problems to report a zero error, which leaves doubts as to the implementation, or possible bias.

Reviewer 3 Report

Authors have addressed all the concerns; therefore in my opinion, the paper can be accepted for the publication.